# *Vista*: A Generalizable Driving World Model with High Fidelity and Versatile Controllability

**Shenyuan Gao**[1,2]    **Jiazhi Yang**[2]    **Li Chen**[2,5]    **Kashyap Chitta**[3,4]    **Yihang Qiu**[2]
**Andreas Geiger**[3,4†]    **Jun Zhang**[1†]    **Hongyang Li**[2,5†]

[1] Hong Kong University of Science and Technology    [2] OpenDriveLab at Shanghai AI Lab
[3] University of Tübingen    [4] Tübingen AI Center    [5] University of Hong Kong

Code and model: `github.com/OpenDriveLab/Vista`
Demo page: `opendrivelab.com/Vista`

## Abstract

World models can foresee the outcomes of different actions, which is of paramount importance for autonomous driving. Nevertheless, existing driving world models still have limitations in generalization to unseen environments, prediction fidelity of critical details, and action controllability for flexible application. In this paper, we present *Vista*, a generalizable driving world model with high fidelity and versatile controllability. Based on a systematic diagnosis of existing methods, we introduce several key ingredients to address these limitations. To accurately predict real-world dynamics at high resolution, we propose two novel losses to promote the learning of moving instances and structural information. We also devise an effective latent replacement approach to inject historical frames as priors for coherent long-horizon rollouts. For action controllability, we incorporate a versatile set of controls from high-level intentions (command, goal point) to low-level maneuvers (trajectory, angle, and speed) through an efficient learning strategy. After large-scale training, the capabilities of Vista can seamlessly generalize to different scenarios. Extensive experiments on multiple datasets show that Vista outperforms the most advanced general-purpose video generator in over $70\%$ of comparisons and surpasses the best-performing driving world model by $55\%$ in FID and $27\%$ in FVD. Moreover, for the first time, we utilize the capacity of Vista itself to establish a generalizable reward for real-world action evaluation without accessing the ground truth actions.

## 1 Introduction

Driven by scalable learning techniques, autonomous driving has made encouraging strides over the past few years [18, 58, 135]. However, intricate and out-of-distribution situations are still intractable for state-of-the-art techniques [83]. One promising solution lies in world models [57, 76], which infer the possible future states of the world from historical observations and alternative actions, in turn assessing the feasibility of such actions. They hold the potential to reason with uncertainty and avoid catastrophic errors [54, 76, 127], thereby promoting generalization and safety in autonomous driving.

Although a primary prospect of world models is to enable the generalization ability to novel environments, existing driving world models are still constrained by data scale [90, 125, 127, 143, 147] and geographical coverage [54, 61]. As summarized in Table 1 and Fig. 1, they are also often confined to low frame rates and resolutions, resulting in a loss of critical details. Furthermore, most models only support a single control modality such as the steering angle and speed. This is insufficient to express various action formats ranging from high-level intentions to low-level maneuvers, and incompatible with the outputs of prevalent planning algorithms [12, 14, 21, 56, 58, 64]. In addition, generalizing action controllability to unseen datasets is understudied. These limitations impede the applicability of existing works, making it imperative to develop a world model that overcomes these limitations.

---

Primary contact to Shenyuan at `sygao@connect.ust.hk`   †Equal advising.

38th Conference on Neural Information Processing Systems (NeurIPS 2024).

Table 1: **Real-world driving world models.** Trained on large-scale high-quality driving data, Vista performs at high spatiotemporal resolution and supports versatile action controllability. `Private data`.

| Method | Model Setups | | | Action Control Modes | | | |
|---|---|---|---|---|---|---|---|
| | Data Scale | Frame Rate | Resolution | Angle&Speed | Trajectory | Command | Goal Point |
| DriveSim [102] | 7h | 5 Hz | 80×160 | ✓ | | | |
| DriveGAN [68] | 160h | 8 Hz | 256×256 | ✓ | | | |
| DriveDreamer [125] | 5h | 12 Hz | 128×192 | ✓ | | | |
| Drive-WM [127] | 5h | 2 Hz | 192×384 | | ✓ | | |
| WoVoGen [90] | 5h | 2 Hz | 256×448 | ✓ | | | |
| ADriver-I [61] | 300h | 2 Hz | 256×512 | | | ✓ | |
| GenAD [136] | 2000h | 2 Hz | 256×448 | | ✓ | ✓ | |
| GAIA-1 [54] | 4700h | 25 Hz | 288×512 | ✓ | | | |
| Vista (Ours) | 1740h | 10 Hz | 576×1024 | ✓ | ✓ | ✓ | ✓ |

Figure 1: **Resolution comparison.** Vista predicts at a higher resolution than previous literature.

To this end, we introduce *Vista*, a driving world model that is proficient in cross-domain generalization, high-fidelity prediction, and multi-modal action controllability. Specifically, we develop the predictive model on a large corpus of worldwide driving videos [136] to foster its generalization ability. To enable coherent future extrapolation, we condition Vista on three essential dynamic priors (Sec. 3.1). Instead of solely relying on the standard diffusion loss [5], we introduce two explicit loss functions to enhance dynamics and preserve structural details (Sec. 3.1), promoting Vista's ability to simulate realistic futures at high resolution. For flexible controllability, we incorporate a versatile set of action formats, including both high-level intentions such as commands and goal points, as well as low-level maneuvers like trajectories, steering angles, and speeds. These action conditions are injected via a unified interface, which is learned through an efficient training strategy (Sec. 3.2). Consequently, as Fig. 2 shows, Vista acquires the ability to anticipate realistic futures at 10 Hz and 576×1024 pixels, and obtains versatile action controllability across various levels of granularity. We also demonstrate the potential of Vista as a generalizable reward function to evaluate the reliability of different actions.

Our contributions are three-fold: **(1)** We present *Vista*, a generalizable driving world model that can predict realistic futures at high spatiotemporal resolution. Its prediction fidelity is greatly improved by two novel losses that capture dynamics and preserve structures, along with exhaustive dynamic priors to sustain consistency in long-horizon rollouts. **(2)** Propelled by an efficient learning strategy, we integrate versatile action controllability into Vista through a unified conditioning interface. The action controllability of Vista can also generalize to different domains in a zero-shot manner. **(3)** We conduct comprehensive experiments across multiple datasets to verify the effectiveness of Vista. It outperforms the most competitive general-purpose video generator and sets a new state-of-the-art on nuScenes. Our empirical evidence shows that Vista can be used as a reward function to assess actions.

## 2 Preliminary

We initialize Vista with the pretrained Stable Video Diffusion (SVD) [5], a latent diffusion model for image-to-video generation. For sampling flexibility, SVD adopts a continuous-timestep formula [66, 111]. It converts data samples $x$ to noise $n$ through a diffusion process $p(n|x) \sim \mathcal{N}(x, \sigma^2 \mathbf{I})$, and generates new samples by progressively denoising the latent towards $\sigma_0 = 0$ from Gaussian noise. The training of SVD can be simplified to minimizing $\mathbb{E}_{x,\sigma,n}\left[\lambda_\sigma \|D_\theta(n; \sigma) - x\|^2\right]$, where $D_\theta$ is a parameterized UNet denoiser and $\lambda_\sigma$ is a re-weighting function omitted hereinafter for brevity. Based on this framework, SVD processes a sequence of noisy latent $n = \{n_1, n_2, ..., n_K\} \in \mathbb{R}^{K \times C \times H \times W}$ and generates a video with $K = 25$ frames. The generation process is guided by a condition image, whose latent is concatenated channel-wise to the inputs, serving as a reference for content generation.

Despite the high aesthetic quality, SVD lacks several key properties to function as a driving world model. As shown in Sec. 4, the first frame predicted by SVD is not identical to the condition image, making it impractical for autoregressive rollout due to content inconsistency. In addition, SVD struggles with the intricate dynamics of driving scenarios, entailing implausible motions. Moreover,

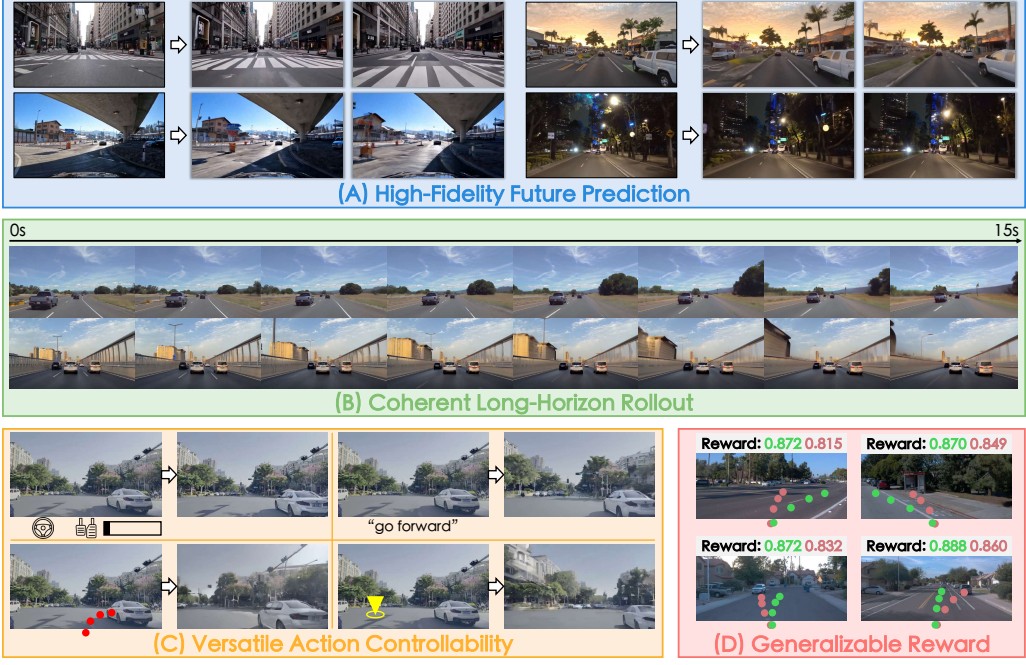

Figure 2: **Capabilities of Vista.** Starting from arbitrary environments, Vista can anticipate realistic and continuous futures at high spatiotemporal resolution **(A-B)**. It can be controlled by multi-modal actions **(C)**, and serve as a generalizable reward function to evaluate real-world driving actions **(D)**.

SVD cannot be controlled by any action format. In contrast, we aim to build a generalizable driving world model that predicts high-fidelity futures with realistic dynamics. It ought to be continuously extendable to long horizons and flexibly controllable by multi-modal actions as illustrated in Fig. 2.

## 3   Learning a Generalizable Driving World Model

As depicted in Fig. 3, Vista adopts a two-phase training pipeline. First, we build a dedicated predictive model, which involves a latent replacement approach to enable coherent future prediction and two novel losses to enhance fidelity (Sec. 3.1). To ensure the generalization to unseen scenarios, we utilize the largest public driving dataset [136] for training. In the second phase, we incorporate multi-modal actions to learn action controllability with an efficient and collaborative training strategy (Sec. 3.2). Using the ability of Vista, we further introduce a generalizable approach to evaluate actions (Sec. 3.3).

### 3.1   Phase One: Learning High-Fidelity Future Prediction

**Basic Setup.** Since world models are initiated to predict futures from the current state, the starting of their prediction should be firmly aligned with the condition image. Therefore, we tailor SVD into a dedicated predictive model by imposing the first frame as the condition image and discarding the noise augmentation [5, 49] during training. With this prediction ability, Vista can perform long-term rollouts by iteratively predicting short-term clips and resetting the condition image with the last clip.

**Dynamic Prior Injection.** Nevertheless, using the aforementioned setup for training often results in irrational dynamics with respect to historical frames, especially in long-term rollouts. We conjecture that this mainly arises from the ambiguity caused by insufficient priors about the tendency of future motions, which is also a common limitation of existing driving world models [54, 68, 125, 127, 136].

Estimating coherent futures requires at least three essential priors that inherently govern the future motion of instances in the scene: position, velocity, and acceleration. Since velocity and acceleration are the first- and second-order derivative of position respectively, these priors can be entirely derived by using three consecutive frames for conditioning. Concretely, we build a frame-wise mask $m \in \{0, 1\}^K$ with a length of $K$ to indicate the presence of condition frames. The mask is set sequentially following the time order, with at most three elements being assigned as 1 to denote three condition

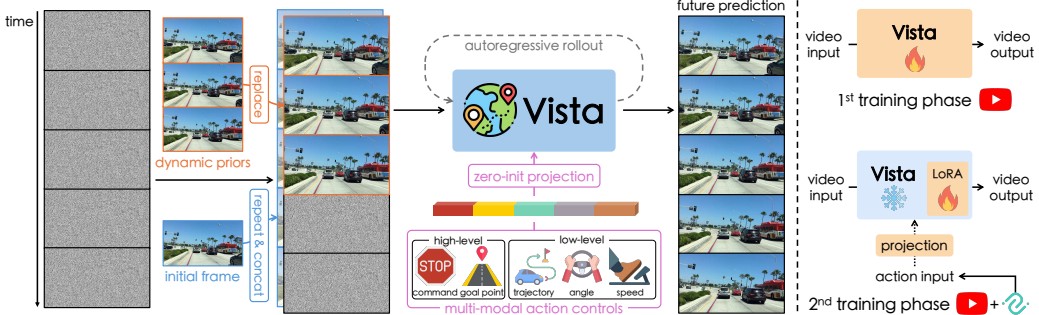

Figure 3: **[Left]: Vista pipeline.** In addition to the initial frame, Vista can absorb more priors about future dynamics via latent replacement. Its prediction can be controlled by different actions and be extended to long horizons through autoregressive rollouts. **[Right]: Training procedure.** Vista takes two training phases, where the second phase freezing the pretrained weights to learn action controls.

frames. Instead of concatenating additional channels to the inputs, we inject new condition frames by replacing the corresponding noisy latent $n_i$ with the clean latent $z_i$ encoded by the image encoder. Formally, the input latent is constructed as $\hat{n} = m \cdot z + (1 - m) \cdot n$ (see Fig. 3 [Left]). To discern the clean latent, we duplicate a new timestep embedding from the pretrained weights and allocate it to the condition frames according to $m$. The timestep embeddings for condition frames and prediction frames are trained separately. Compared to channel-wise concatenation, we find that replacing the latent is more effective and flexible in absorbing varying numbers of condition frames. In addition, we observe that the replaced latent, when applied to SVD directly, does not degrade its generation quality. Thus, the original performance will not be disturbed when the training is launched. Since there is no need to predict the observed condition frames, we exclude them from the loss as follows:

$$\mathcal{L}_{\text{diffusion}} = \mathbb{E}_{z,\sigma,\hat{n}} \Big[ \sum_{i=1}^{K} (1 - m_i) \odot \| D_\theta(\hat{n}_i; \sigma) - z_i \|^2 \Big], \tag{1}$$

where $D_\theta$ is the UNet denoiser that shares the same architecture with SVD. With the replaced latent holding sufficient priors, Vista can fully capture the status of the surrounding instances and predict more coherent and plausible long-term futures through iterative rollouts. In practice, we leverage the last three frames of a predicted clip as dynamic priors for the next prediction step during rollouts.

**Dynamics Enhancement Loss.** Unlike general videos that cover rather small spaces, driving videos capture much larger scenes [136]. In most driving videos, distant and monotonous regions dominate the view, with the moving foreground instances only occupying a relatively small area [17]. However, the latter often exhibit higher stochasticity, complicating their prediction. Since Eq. (1) supervises all outputs uniformly, it cannot effectively discriminate the nuances of different regions as Fig. 4(b) shows. As a result, the model cannot efficiently learn to predict realistic dynamics in crucial regions.

As the discrepancy between two adjacent frames provides considerable motion patterns [123, 132], we introduce an additional supervision to encourage the learning of dynamics for crucial regions. To be specific, we first introduce a dynamics-aware weight $w = \{w_2, w_3, ..., w_k\} \in \mathbb{R}^{K-1 \times C \times H \times W}$ that highlights the regions where the prediction has inconsistent motion compared to the ground truth:

$$w_i = \| (D_\theta(\hat{n}_i; \sigma) - D_\theta(\hat{n}_{i-1}; \sigma)) - (z_i - z_{i-1}) \|^2. \tag{2}$$

For numerical stability, we normalize $w$ within each video clip. As shown in Fig. 4(c), the weight amplifies the presence of large motion disparities, highlighting dynamic regions while excluding monotonous backgrounds. Given the causality of future prediction, *i.e.* subsequent frames ought to follow previous ones, we define a new loss by penalizing the latter frame of each adjacent frame pair:

$$\mathcal{L}_{\text{dynamics}} = \mathbb{E}_{z,\sigma,\hat{n}} \Big[ \sum_{i=2}^{K} \mathbf{sg}(w_i) \odot (1 - m_i) \odot \| D_\theta(\hat{n}_i; \sigma) - z_i \|^2 \Big], \tag{3}$$

where $\mathbf{sg}(\cdot)$ stops the gradient. By adaptively re-weighting the standard diffusion loss, $\mathcal{L}_{\text{dynamics}}$ can boost the learning efficiency of dynamic regions, *e.g.*, the moving vehicles and sidewalks in Fig. 4(d).

**Structure Preservation Loss.** The trade-off between perceptual quality and motion intensity has been widely acknowledged in video generation [3, 32, 73, 144], and our case is no exception. When

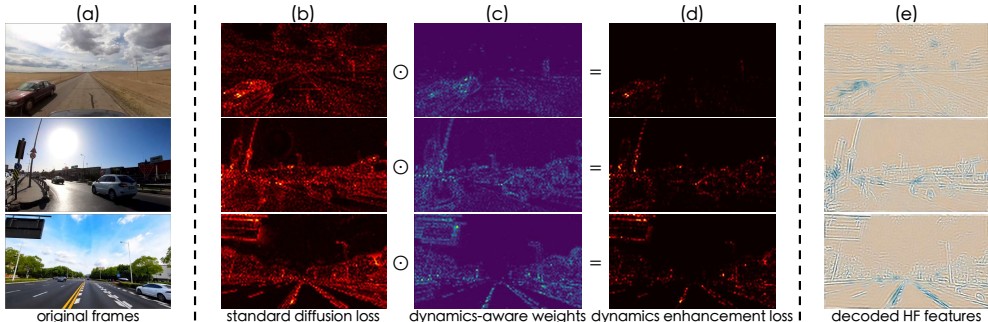

Figure 4: **Illustration on loss design.** Different from the standard diffusion loss **(b)** that is distributed uniformly, our dynamics enhancement loss **(d)** enables an adaptive concentration on critical regions **(c)** (*e.g.*, moving vehicles and roadsides) for dynamics modeling. Moreover, by explicitly supervising high-frequency features **(e)**, the learning of structural details (*e.g.*, edges and lanes) can be enhanced.

it comes to high-resolution prediction for dynamic driving scenarios, we discover that the predicted structural details degrade severely with over-smoothed or broken objects, *e.g.*, the outlines of vehicles unravel quickly as they move (see Fig. 12). To alleviate this problem, it is important to place more emphasis on structural details. Based on the fact that structural details, such as edges and textures, mainly reside in high-frequency components, we identify them in the frequency domain as follows:

$$z'_i = \mathcal{F}(z_i) = \texttt{IFFT}\big(\mathcal{H} \odot \texttt{FFT}(z_i)\big), \qquad (4)$$

where `FFT` and `IFFT` are the 2D discrete Fourier transform and inverse discrete Fourier transform respectively, and $\mathcal{H}$ is an ideal 2D high-pass filter that truncates low-frequency components under a certain threshold. The Fourier transforms are applied on each channel of $z_i$ independently. As illustrated in Fig. 4(e), features associated with structural information can be effectively emphasized by Eq. (4). The corresponding features from the predicted latent $D_\theta(\hat{n}_i; \sigma)$ can also be extracted similarly. With the extracted high-frequency features, we devise a new structure preservation loss as:

$$\mathcal{L}_{\text{structure}} = \mathbb{E}_{\boldsymbol{z},\sigma,\hat{\boldsymbol{n}}}\Big[ \sum_{i=1}^{K}(1 - m_i) \odot \|\mathcal{F}(D_\theta(\hat{n}_i; \sigma)) - \mathcal{F}(z_i)\|^2\Big]. \qquad (5)$$

This loss function minimizes the disparity of high-frequency features between prediction and ground truth, so that more structural information can be retained. Our final training objective is a weighted sum of Eq. (1), Eq. (3) and Eq. (5), where $\lambda_1$ and $\lambda_2$ are trade-off weights to balance the optimization:

$$\mathcal{L}_{\text{final}} = \mathcal{L}_{\text{diffusion}} + \lambda_1\mathcal{L}_{\text{dynamics}} + \lambda_2\mathcal{L}_{\text{structure}}. \qquad (6)$$

### 3.2 Phase Two: Learning Versatile Action Controllability

**Unified Conditioning of Versatile Actions.** To maximize usage flexibility, a driving world model should be able to leverage multiple action formats with different characteristics. For instance, one may use the world model to evalute high-level policies [127], or to execute low-level maneuvers [102]. However, existing approaches only support limited action controls [54, 61, 90, 125, 127], inhibiting their flexibility and applicability. Therefore, we incorporate a versatile set of action modes for Vista: **(1) Angle and Speed** stand for the utmost fine-grained action controls. We normalize angles to $[-1, 1]$ and represent speeds in $km/h$. The signals from different timestamps are concatenated sequentially. **(2) Trajectory** is a series of 2D displacements in ego coordinates. It is widely used as the output of planning algorithms [12, 21, 58, 62, 63]. We represent the trajectory in meters and flatten it into a sequence. **(3) Command** is the most high-level intention. Without loss of generality, we define four commands, *i.e.* go forward, turn right, turn left, and stop, which are implemented as categorical indices. **(4) Goal Point** is a 2D coordinate projected from the short-term ego destination onto the initial frame, serving as an interactive interface [74]. The coordinate is normalized by the image size.

Note that these actions are heterogeneous and cannot be used interchangeably. After transforming all these actions into numerical sequences, we encode them as a unified concatenation of Fourier embeddings [114, 116] (see Fig. 3). These embeddings can be jointly ingested by learning additional projections to expand the input dimension of the cross-attention layers in the UNet [5]. The new

projections are initialized as zeros to enable gradual learning from the pretrained state. We empirically discover that incorporating action conditions through cross-attention layers yields faster convergence and stronger controllability compared to other approaches such as additive embeddings [128, 136].

**Efficient Learning.** We learn action controllability after the first training phase. Since the number of total iterations is crucial for diffusion training [5, 22, 32, 99], we separate action control learning into two stages. In the first stage, we train our model at a low resolution (320×576), which achieves 3.5× higher training throughput compared to the original resolution (576×1024). This stage constitutes the majority of training iterations. Then, we finetune the model at the desired resolution (576×1024) for a short duration, so that the learned controllability can effectively cater to high-resolution prediction.

However, tuning the UNet [5] at a lower resolution directly may undermine the high-fidelity prediction ability. Conversely, freezing all UNet weights and training the new projections alone would precipitate a quality decline (see Appendix D), suggesting the necessity to make the UNet adaptable. To solve this, we freeze the pretrained UNet and introduce parameter-efficient LoRA adapters [55] to each attention layer. After training, the low-rank matrices can be seamlessly integrated with the frozen weights, without introducing extra inference latency. Thus, the pretrained weights remain intact when training at the low resolution, avoiding deterioration of the pretrained high-fidelity prediction ability.

Since the parameters of the camera and vehicle are unavailable for open-world scenarios, it seems impossible to obtain multiple equivalent action conditions simultaneously at inference time. Additionally, it will entail prohibitively expensive training to encompass all possible combinations of action conditions. Hence, unlike common practices that activate all conditions during training, we enforce the independence of different action formats by enabling only one of them for each training sample. The remaining action conditions will be filled with zeros as unconditional inputs. As demonstrated in Appendix D, this simple constraint prevents the squandering of training cost on action combinations and maximizes the learning efficiency of each individual action mode within the same training steps.

**Collaborative Training.** Note that the aforementioned action conditions are not available in OpenDV-YouTube [136]. On the other hand, nuScenes [10] has adequate annotations to derive these conditions. To maintain generalization and learn controllability in tandem, we introduce a collaborative training strategy by utilizing the samples from both datasets, with the action conditions for OpenDV-YouTube set to zero. The action control learning phase adopts the same loss as Eq. (6). By learning from two complementary datasets, Vista gains versatile controllability that are generalizable to novel datasets.

### 3.3 Generalizable Reward Function

One application of world models is to evaluate actions by engaging a reward module [40, 42, 43, 76]. Drive-WM [127] establishes a reward using external detectors [82, 84]. However, these detectors are developed on a particular dataset [10], which may become a bottleneck for reward estimation in arbitrary scenarios. On the other hand, Vista has ingested millions of human driving logs, exhibiting strong generalization across scenes. Based on the observation that out-of-distribution conditions will lead to increased diversity in generation [28, 60], we utilize the prediction uncertainty from Vista itself as the source of our reward. Different from Drive-WM, our reward function seamlessly inherits the generalization of Vista without resorting to external models. Specifically, we estimate uncertainty via conditional variance. For reliable approximation, we denoise from randomly sampled noise with the same condition frame $c$ and action $a$ for $M$ rounds. Our reward function $R(c, a)$ is then defined as the exponential of averaged negative conditional variance:

$$\mu' = \frac{1}{M} \sum_m D_\theta^{(m)}(\hat{n}; c, a), \tag{7}$$

$$R(c, a) = \exp\left[\mathrm{avg}\left(-\frac{1}{M-1} \sum_m (D_\theta^{(m)}(\hat{n}; c, a) - \mu')^2\right)\right], \tag{8}$$

where $\mathrm{avg}(\cdot)$ averages all latent values within the video clip. Based on this formulation, unfavorable actions with larger uncertainties will lead to lower rewards. In contrast to commonly used evaluation protocols (*e.g.*, the L2 error), our reward function can evaluate actions without referring to the ground truth actions. Note that we do not normalize the estimated rewards for the simplicity of definition, but it is straightforward to amplify the relative contrast by rescaling the estimated rewards with a factor.

Table 2: **Comparison of prediction fidelity on nuScenes validation set.** Vista achieves encouraging results that outperform the state-of-the-art driving world models with a significant performance gain.

| Metric | DriveGAN [102] | DriveDreamer [125] | WoVoGen [90] | Drive-WM [127] | GenAD [136] | Vista (Ours) |
|---|---|---|---|---|---|---|
| FID ↓ | 73.4 | 52.6 | 27.6 | 15.8 | 15.4 | **6.9** |
| FVD ↓ | 502.3 | 452.0 | 417.7 | 122.7 | 184.0 | **89.4** |

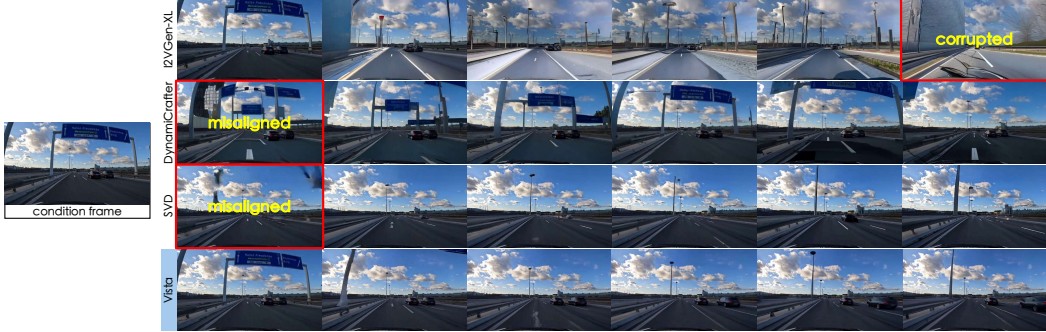

Figure 5: **Driving futures predicted by different models using the same condition frame**. We contrast Vista to publicly available video generation models using their default configurations. Whilst previous models produce misaligned and corrupted results, Vista does not suffer from these caveats.

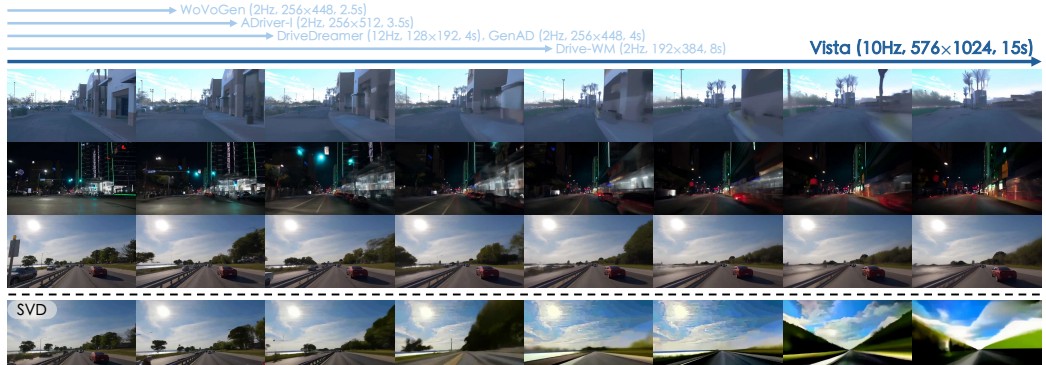

Figure 6: **[Top]: Long-horizon prediction.** Vista can forecast 15 seconds high-resolution futures without much degradation, encompassing long driving distances. The length of the blue lines indicate the duration of the longest prediction showcased by previous works. **[Bottom]: Long-term extension results of SVD.** SVD fails to generate consistent high-fidelity videos autoregressively as Vista does.

## 4 Experiments

In this section, we first demonstrate Vista's strengths in generalization and fidelity in Sec. 4.1. We then show the impact of action controls in Sec. 4.2. We also substantiate the efficacy of the proposed reward function in Sec. 4.3. Finally, we conduct ablation studies on our key designs in Sec. 4.4. For more implementation details and experimental results, please refer to Appendix C and Appendix D.

### 4.1 Comparisons of Generalization and Fidelity

**Automatic Evaluation.** Since none of the driving world models are publicly accessible, we compare these methods with their quantitative results on nuScenes. We filter 5369 valid samples from the validation set to conduct FID [47] and FVD [115] evaluation. For FID evaluation, we crop and resize the predicted frames to the resolution of 256×448. For FVD evaluation, we use all 25 frames in each video clip and downsample them to 224×224 following LVDM [46]. Table 2 reports the results of all methods. In both metrics, Vista surpasses previous driving world models with a considerable margin.

**Human Evaluation.** To analyze the generalization of Vista across different datasets, we compare it against three prominent general-purpose video generators trained on web-scale data [5, 133, 144] (see Fig. 5). It is known that automatic metrics like FVD [115] cannot conclusively reveal perceptual

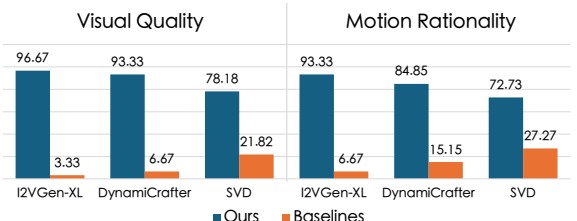

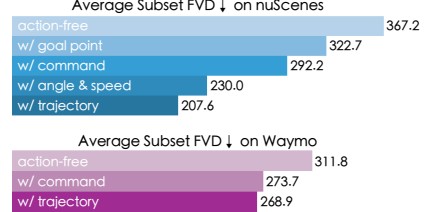

Figure 7: **Human evaluation results.** The value denotes the percentage of the times that one model is preferred over the other. Vista outperforms existing works in both metrics.

Figure 8: **Efficacy of action controls.** Applying action controls will produce more similar predictions to the real data.

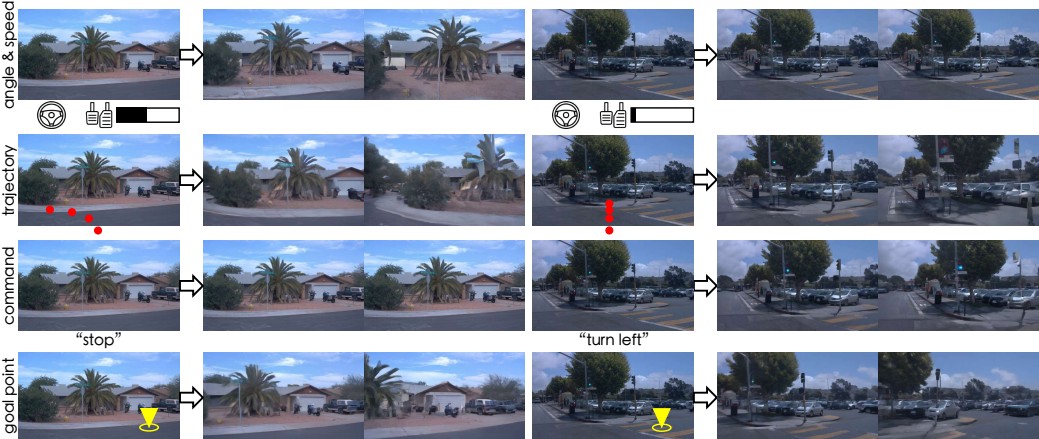

Figure 9: **Versatile action controllability.** Vista can predict the corresponding outcomes in response to multi-modal action conditions across diverse scenarios. More results can be found in Appendix E.

quality [3, 6, 32, 130, 136], let alone real-world dynamics. Therefore, we opt for human evaluation for more faithful analysis. Following recent advances [3, 5, 6, 15, 16, 32, 122, 126], we adopt the Two-Alternative Forced Choice protocol. Specifically, participants are presented with a side-by-side video pair and asked to choose the video they deem better on two orthogonal aspects: visual quality and motion rationality. To avoid potential bias, we crop each video to a fixed aspect ratio, downsample them to the same resolution, and trim the excess frames when Vista generates longer videos than others. We only feed one condition frame to align with other models. To ensure the variety of scenes, we uniformly assemble 60 scenes from four representative datasets: OpenDV-YouTube-val [136], nuScenes [10], Waymo [112], and CODA [79]. These datasets collectively exemplify the intricacy and diversity of real-world driving, *e.g.*, OpenDV-YouTube-val includes geofenced districts, Waymo offers a unique domain compared to our training data, and CODA contains extremely challenging corner cases. We collect a total of 2640 answers from 33 participants. As presented in Fig. 7, Vista outperforms all baselines on both aspects, demonstrating its profound comprehension of the driving dynamics. Further, unlike other models that are only applicable for short-term generation, Vista can accommodate more dynamic priors and produce coherent long-horizon rollouts as shown in Fig. 6.

## 4.2 Results of Action Controllability

**Quantitative Results.** To evaluate the impact of action controls, we divide the validation set of both nuScenes and the unseen Waymo dataset into four subsets according to our command categories. We then generate predictions using different modalities of the ground truth actions. The FVD score is measured on each subset and then averaged. A lower FVD score reflects a closer distribution to the ground truth videos, indicating that the predictions exhibit more resemblance to each specific type of behavior. Fig. 8 shows that our action controls can emulate the corresponding movements effectively.

We also introduce a new metric named *Trajectory Difference* to assess control consistency. Following GenAD [136], we train an inverse dynamics model (IDM) that estimates the corresponding trajectory from a video clip. An illustration of IDM is shown in Fig. 13. We then send Vista's prediction to the IDM and calculate the L2 difference between the ground truth trajectory and the estimated trajectory.

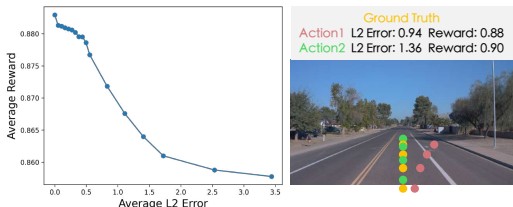 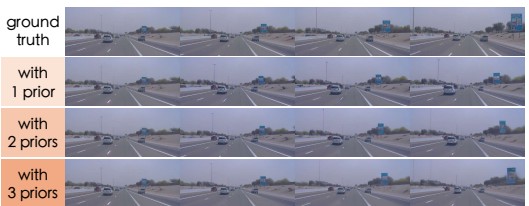

Figure 10: **[Left]: Average reward on Waymo with different L2 errors. [Right]: Case study.** The relative contrast of our reward can properly assess the actions that the L2 error fails to judge.

Figure 11: **Effect of dynamic priors.** Injecting more dynamic priors yields more consistent future motions with the ground truth, such as the motions of the white vehicle and the billboard on the left.

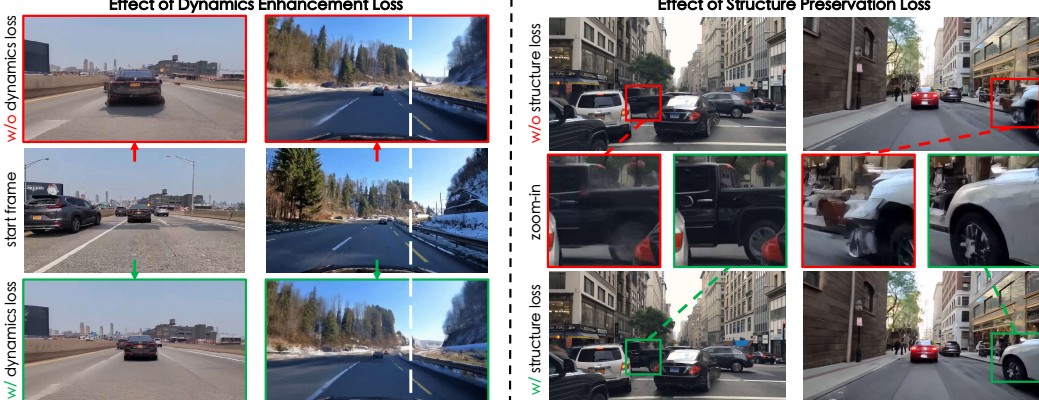

Figure 12: **[Left]: Effect of dynamics enhancement loss.** The model supervised by the dynamics enhancement loss generates more realistic dynamics. In the first example, instead of remaining static, the front car moves forward normally. In the second example, when the ego-vehicle steers right, the trees shift towards the left naturally adhering to the real-world geometric rules. **[Right]: Effect of structure preservation loss.** The proposed loss yields a clearer outline of the objects as they move.

The differences are measured over 2 seconds. The lower the trajectory difference, the stronger the control consistency Vista exhibits. We conduct the experiments on nuScenes and Waymo. For each dataset, we collect a subset that contains 537 samples. As reported in Table 3, Vista can be effectively controlled by different modalities of actions, resulting in more consistent motions to the ground truth.

**Qualitative Results.** Fig. 9 exhibits the versatile action controllability of our model. Vista can be effectively controlled by multi-modal actions, even in unseen scenarios beyond the training domain. In Appendix E, we also showcase the counterfactual reasoning ability of Vista using abnormal actions.

### 4.3 Results of Reward Modeling

To validate the efficacy of our reward function, we jitter the ground truth trajectories into a series of inferior trajectories. Specifically, we compute the standard deviation of each waypoint from the nuScenes training set as prior distributions. These priors are jointly rescaled to sample perturbations with different L2 errors. The perturbations are then added as offsets to the ground truth trajectories. To ensure the plausibility of sampled trajectories, we adopt an explicit correlating strategy [35, 95] to regularize offset sampling and recursively sample new trajectories until their offsets are consistent in tendencies. To demonstrate the generality of our reward function, we conduct reward estimation on Waymo [112], which is unseen in training. This is done by uniformly sampling from each command category on Waymo validation set, resulting in 1500 cases in total. We compare the average reward of the trajectories with varying L2 errors in Fig. 10. Our reward decreases when the deviation from the ground truth increases, underscoring the potential of our approach to serve as a viable reward function. It also holds the promise to remedy the irrationality in current evaluation protocols for planning [18, 83, 141], such as the L2 error shown in Fig. 10. More in-depth analysis of rewards, including sensitivity to hyperparameters and reward of other actions, are provided in Appendix D.

Table 3: **Impacts of different action conditions and dynamic priors.** By applying action conditions and dynamic priors, Vista can predict motion that is more consistent compared to the ground truth.

| Dataset | Condition | Average Trajectory Difference ↓ | | |
|---------|-----------|:---:|:---:|:---:|
| | | with 1 prior | with 2 priors | with 3 priors |
| nuScenes | GT video | 0.379 | 0.379 | 0.379 |
| | action-free | 3.785 | 2.597 | 1.820 |
| | + goal point | 2.869 | 2.192 | 1.585 |
| | + command | 3.129 | 2.403 | 1.593 |
| | + angle & speed | 1.562 | 1.123 | 0.832 |
| | + trajectory | 1.559 | 1.148 | 0.835 |
| Waymo | GT video | 0.893 | 0.893 | 0.893 |
| | action-free | 3.646 | 2.901 | 2.052 |
| | + command | 3.160 | 2.561 | 1.902 |
| | + trajectory | 1.187 | 1.147 | 1.140 |

## 4.4 Ablation Study

**Dynamic Priors.** Fig. 11 visualizes the outcomes of using different orders of dynamic priors. The order of priors corresponds to the number of condition frames. It shows that dynamic priors play a pivotal role in long-horizon rollouts, where the coherence with respect to historical frames is essential.

To further demonstrate the efficacy of dynamic priors, we conduct a quantitative evaluation in Table 3. Specifically, we use the IDM in Sec. 4.2 to infer the trajectories of the predicted videos with different orders of dynamic priors. The diminishing differences in trajectory suggest that introducing more priors can effectively improve the consistency between prediction and ground truth.

**Auxiliary Supervisions.** To verify the effectiveness of the two losses proposed in Sec. 3.1, we devise two additional variants by individually ablating each loss from a variant that incorporates both losses. We qualitatively compare their effects through Fig. 12, which confirms that the dynamics enhancement loss can promote the learning of real-world dynamics, and the structure preservation loss can reinforce the prediction of structural details.

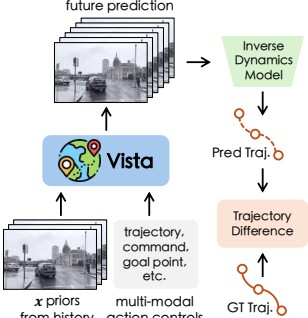

Figure 13: **An illustration of the IDM experiments in Table 3.**

## 5 Conclusion

In this paper, we introduce *Vista*, a generalizable driving world model with enhanced fidelity and controllability. Based on our systematic investigations, Vista is able to predict realistic and continuous futures at high spatiotemporal resolution. It also possesses versatile action controllability that is generalizable to unseen scenarios. Moreover, it can be formulated as a reward function to evaluate actions. We hope Vista will usher in broader interest in developing generalizable autonomy systems.

**Limitations and future work.** As an early endeavor, Vista still exhibits some limitations with respect to computation efficiency, quality maintenance, and training scale. Our future work will look into applying our method to scalable architectures [54, 97]. More discussions are included in Appendix A.

## Acknowledgments

This work is supported by National Key R&D Program of China (2022ZD0160104), National Natural Science Foundation of China (62206172), and Shanghai Committee of Science and Technology (23YF1462000). This work is also partially supported by the BMBF (Tübingen AI Center, FKZ: 01IS18039A), the DFG (SFB 1233, TP 17, project number: 276693517), and the EXC (number 2064/1 – project number: 390727645). We thank the International Max Planck Research School for Intelligent Systems (IMPRS-IS) for supporting Kashyap Chitta. We also appreciate Zetong Yang, Chonghao Sima, Linyan Huang, and the rest members from OpenDriveLab for valuable feedback. We express our sincere gratitude to all anonymous participants for helping with the human evaluation.

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

# Appendix

# A Discussions

To help a thorough understanding of this work, we discuss intuitive questions that might be raised.

**Q1.** *Why is at least position, velocity, and acceleration required to predict coherent futures?*

Position ensures the predicted future begins continuously with the current state. Velocity manifests how objects are moving, *e.g*., whether they are turning left or turning right. Acceleration represents how velocity changes over time, *e.g*., whether the surroundings are moving faster or moving slower. Without utilizing acceleration as a cue, a car overtaking the ego-vehicle may suddenly be passed by in the next autoregressive prediction step. These three priors provide essential cues to allow consistent future extension with respect to historical observation.

**Q2.** *How is the form of the proposed reward function defined?*

Unlike VIPER [28] and Diffusion Reward [60] that both make discrete predictions, our model predicts continuous latent. Therefore, our reward is estimated according to conditional variance rather than log-likelihood or entropy. In addition, measuring uncertainty with log-likelihood requires comparing the prediction to the ground truth. As we deploy the reward in any scenario, the approach of VIPER is infeasible for our objective. Note that our reward calculation is meticulously designed to satisfy the Kolmogorov axioms, *i.e.* it is non-negative and the measure of the entire sample space is $[0, 1]$.

**Q3.** *Reward estimation efficiency compared to the detector-based method* [127].

Though our reward estimation involves multi-round denoising, it will not spend more compute than the detector-based reward function defined in Drive-WM [127]. To be specific, Drive-WM obtains the rewards from the perception results. Given that the detectors [82, 84] take image sequences as inputs, Drive-WM has to accomplish all steps of the denoising process before perception. Differently, our reward function estimates the reward with the uncertainty that originates from the world model itself without relying on other perception models. Therefore, the estimation of uncertainty does not require completing the generation process. It can be realized by only denoising each sample for a few steps. In fact, as specified in Appendix C, the total computation required for reward estimation per situation (10 steps, 5 rounds) is no greater than that of generating the entire video (50 steps for our model) as Drive-WM does. Note that the computational cost for our reward estimation can be flexibly reduced to further improve its efficiency. As shown in Fig. 14, using 5 denoising steps (50% of the default computation) also yields satisfactory estimations of the reward.

**Q4.** *Usage of the proposed reward function.*

**(1)** As mentioned in Sec. 4.3, the proposed reward function can potentially serve as an alternative metric of driving actions that mitigate the concerns in existing open-loop evaluation [18, 83, 141]. **(2)** As demonstrated in Fig. 10, better actions generally yield higher rewards with our reward function. Taking advantage of this property, there is great promise for our reward function to be used as a critic module [76], which enables model-predictive control by executing the optimal action that maximizes the estimated reward [27, 30, 35]. This procedure can be performed in conjunction with distribution-based planners [53, 148] that can make action proposals to reduce the searching space.

**Q5.** *Any other potential applications of Vista?*

**(1)** As a generalizable predictive model, Vista could be utilized as a forward dynamics model [13, 26] to simulate short-term dynamics and assist long-horizon planning tasks like visual navigation [106]. **(2)** It is also intriguing to utilize Vista as an implicit driving policy, which is spontaneously acquired through future prediction [1, 25]. After synthesizing the video plan, we can convert the resultant image trajectory to executable actions by a non-causal inverse dynamics model [4, 65, 71], which can be efficiently learned from much fewer data compared with the imitation learning pipeline [2, 9]. In autonomous driving, the inverse dynamics model could be implemented with visual odometry [75]. **(3)** In collaboration with the reward function, it is also worth investigating if Vista could facilitate model-based reinforcement learning by boosting the sampling efficiency in real-world scenarios [137].

**Q6.** *Differences with GenAD* [136].

These two works have fundamental differences in control versatility and prediction fidelity. First of all, Vista is a generalizable world model that can be controlled by multi-modal action conditions. Although GenAD has also trained a trajectory-conditioned extension, its weights are fully finetuned on nuScenes and the generalization of its action control has never been validated. In contrast, Vista integrates versatile action controllability that can generalize to new scenarios in a zero-shot manner. Unlike GenAD that requires labeling OpenDV-YouTube with commands and texts, our collaborative training strategy skillfully averts this labor that may incur accumulated noises and conflicts [136]. In addition, Vista (10 Hz, 576×1024) operates at much higher frame rate and resolution, considerably beyond the capability of GenAD (2 Hz, 256×448) in both temporal and spatial axes. Different from GenAD, we also put forth several dedicated designs for high-fidelity prediction. We find that Vista, with a lower model complexity, achieves much better FID and FVD scores than GenAD (see Table 2).

**Q7.** *Limitations, failure cases, and possible solutions.*

As one of the pioneering efforts, Vista still has a few limitations that call for future works. **(1)** Since Vista predicts futures at an exceptional spatiotemporal resolution, it is inevitable to be computationally expensive, particularly in downstream applications. Potential solutions may include faster sampling techniques [89, 146] and training-based distillations [91, 93, 101, 103, 110, 124]. **(2)** It is possible that the prediction may undergo an apparent degradation in long-horizon rollouts or drastic view shifts. Extra refinements on the prediction results [6, 49, 54, 99, 126, 144] could be helpful. Speculatively, applying our recipe to more scalable architecture [54, 97] is also promising to address this limitation. **(3)** Similar to other controllable video generation methods [128], the chance of failure still persists in our action controls, especially for ambiguous intentions such as commands and goal points as Fig. 8 reveals. Incorporating more datasets with action annotations [11, 112, 129] for collaborative training could be beneficial. Using compositional classifier-free guidance [8, 19, 32] to amplify the individual impact of action conditions may also help (at a cost of increased inference compute). **(4)** Although our training data is based on the largest public driving dataset [136], it is nowhere near the entirety of Internet driving data, thus leaving huge untapped potential to further expand the capabilities of Vista.

**Q8.** *Why not expand the Vista framework to surround-view generation?*

It is true that supporting surround-view generation would further help driving. Existing works [127, 145] have explored the surround-view settings on nuScenes [10]. However, in this paper, we focus on the front-view setting for three main reasons: **(1)** The front view setting allows leveraging diverse data sources [54, 136]. Conversely, the distinctions in multi-view videos from various datasets, such as different numbers of cameras, hinder unified modeling and data scaling. **(2)** Models that focus on the front view can be seamlessly applied to different datasets without adaptation [107], broadening their applicability across datasets. **(3)** Though incomplete, the front view often contains most of the information vital for driving. As demonstrated in NAVSIM [23], using the front view alone results in only a $1.1\%$ performance downtick in collision rate compared to using five surround-view cameras.

**Q9.** *Broader impact.*

Despite the encouraging improvements, our work is by no means perfect when it comes to real-world applications that involve dealing with highly complicated situations. As Vista is based on the diffusion framework, which introduces stochastic outcomes and non-negligible latencies, deploying it into autonomous vehicles directly could pose safety risks. While it is not a silver bullet yet, we expect that Vista will inspire the community to further exploit the capabilities and applications of driving world models. As a prototype for generalizable driving world models, we hope that Vista can stimulate the investigations in developing generalizable systems for autonomous driving and machine intelligence.

## B  Related Work

### B.1  World Models

Intelligent agents should be able to make efficacious decisions even under unseen situations [9, 57, 76, 113, 138, 149]. This requires fundamental knowledge of the world that generalizes to rare cases. As an

internal manifestation of such knowledge, a world model predicts plausible futures of the world given potential actions [9, 40, 69, 76, 96, 113, 137]. In principle, it not only predicts how the environment will unfold over time, but also deduces the underlying physical dynamics and agentic behaviors. Such properties can be useful for representation learning [35, 45, 88, 105, 131], model-based reinforcement learning [39, 40, 42, 43, 94, 96, 98], as well as model-predictive control [27, 30, 35, 41, 92, 104, 142]. Recent methods [37, 44, 80, 85, 86] also induce language-based world models from large language models, but are restricted in textual space and struggle with grounding on physics [26, 59].

Although world models have been extensively applied and made significant revolutions in simulated games [40, 42, 43] and indoor embodiment [72, 92, 120], such investigations for autonomous driving still lag behind [127, 143]. Different from other tasks, world modeling for autonomous driving poses unique challenges, which primarily arise from the large field of views with highly dynamic motions. Some practices imagine the world in the bird's eye view (BEV) space [20, 31, 52, 53, 78, 81]. Recent practices model the world state as raw sensor observations such as point clouds [7, 67, 140, 143, 147] and images [54, 61, 68, 90, 102, 125, 127, 131, 145]. The latter category, namely visual world models, hold more promise for scaling up due to sensor flexibility and data accessibility. Nevertheless, existing methods are restricted to a particular dataset [61, 77, 90, 125, 127, 143, 145, 147] or simulator [7, 131], compromising their generalization ability to novel domains. Meanwhile, these efforts lack systematic designs for the driving domain and only model the world at relatively low frame rates and resolutions, which discards the fine-grained details and impairs their ability to express real-world behaviors. Moreover, most of them are restricted to a specific control modality [54, 61, 90, 125], which hinders the accommodation to prevailing planning algorithms [12, 14, 21, 56, 58, 64] and extension to more applications like decision-making [127] or user interaction [74]. Besides, existing methods seldom explore zero-shot action controllability across different datasets. The inferior generalization, fidelity and controllability collectively preclude existing driving world models from broadly facilitating the development of autonomous driving.

## B.2 Video Generation

Video generation is an effective way to model the world and has undergone remarkable advancements over the years. Pioneering works [119, 134] have studied various kinds of generative models [29, 33, 70]. Swayed by the success of diffusion models [24, 48, 100], a surge of diffusion-based video generation methods have emerged [6, 34, 46, 51, 108, 117, 121]. Recent works [5, 15, 32, 144] shift their focus towards image-to-video generation for its finer content description and better scalability in training data. However, most of them are not strict predictive models that generate videos starting from the condition image. Moreover, existing methods struggle with the intricate dynamics in driving scenarios from the ego perspective [136], which limits their feasibility as driving world models.

While the majority of existing methods produce videos without explicit controllability, two recent works [128, 139] introduce camera motion control to video generation. However, camera motion is conceptually distinct from vehicle actions and both of these works are text-to-video methods without any prediction ability. Contrarily, the model we developed is a predictive world model that produces realistic dynamics and allows versatile action controls for autonomous driving.

## C Implementation Details

### C.1 Model

We adopt the framework of SVD [5] as the architecture of Vista, which consists of 2.5B parameters in total, including 1.6B UNet parameters. For action conditioning, we encode the value of each action sequence into Fourier embeddings [114, 116] with 128 channels.

### C.2 Dataset

We utilize a rigorously filtered set of OpenDV-YouTube [136] for training, and incorporate nuScenes training set [10] during the action control learning phase. Concretely, we manually eliminate 15 hours of irrelevant content from OpenDV-YouTube, yielding approximately 1735 hours of unlabeled driving videos. Since nuScenes is heavily biased [83, 107, 127], we balance its samples based on command categories to foster the learning of rare actions. The video clips are sampled with 25 frames at 10 Hz. Although nuScenes [10] is logged at 12 Hz, we find no negative impact of treating them

as 10 Hz videos. The model inputs are composed by cropping and resizing these clips to the target resolution. We define an action as a sequence comprising 25 frames. To categorize actions into commands, we follow the established conventions in planning [56, 58, 64] and define the command of ego-vehicle as `"turn right"` or `"turn left"` when its final displacement exceeds 2 meters in the orthogonal direction relative to its initial heading. To allow more precise categorization, we additionally introduce a `"stop"` command when the forward driving distance is less than 2 meters.

### C.3 Training

At the first training phase, we train all UNet parameters at $576 \times 1024$ resolution on 128 A100 GPUs for 20K iterations, which takes about 8 days in total. We accumulate the gradients of 2 steps, yielding an effective batch size of 256. Following SVD, our model is trained with the EDM framework [66]. We use the AdamW optimizer [87] with a learning rate of $1 \times 10^{-5}$. The learning rate for spatial layers is moderated by a discount factor of 0.1. The coefficients $\lambda_1$ and $\lambda_2$ in Eq. (6) are set to 1.0 and 0.1 respectively. Offset noise [38] is also used with a strength of 0.02 as it helps improve temporal smoothness. We randomly sample different orders of dynamic priors with increasing probabilities, *i.e.* $1/15$, $2/15$, $4/15$, $8/15$ for 0, 1, 2, 3 condition frames respectively. The noise augmentation [49] is disabled to retain more details from the condition frames.

As for the action control learning phase, we freeze the pretrained weights and add LoRA [55] and projection layers to all attention blocks of the UNet. The rank of LoRA is set to 16. We then train the new weights at $320 \times 576$ resolution for 120K iterations using batch size 8 and learning rate $5 \times 10^{-5}$. After the controllability can be clearly witnessed, we continue to finetune the unfrozen weights at $576 \times 1024$ resolution for another 10K iterations. We drop out each activated action mode with a ratio of 15% to allow classifier-free guidance [50]. The sampling ratio of OpenDV-YouTube and nuScenes is $1 : 1$ at this training phase. The whole training process for action controllability takes around 10 days on 8 A100 GPUs, with roughly 8 days at the low resolution and 2 days at the high resolution.

### C.4 Sampling

We generate future videos using the DDIM sampler [109] for 50 steps. The sampling starts with $\sigma_{\max}$ at 700.0. Since our model predicts long-term futures in an autoregressive manner, the issue of over-saturation caused by the standard classifier-free guidance will accumulate rapidly. Therefore, unlike SVD that linearly increases the guidance scale, we employ a triangular classifier-free guidance scheme [118] to permit genuine long-horizon rollouts. Specifically, for the $i$-th frame in each $K$ frames to predict, we assign its guidance scale $s(i)$ following:

$$s(i) = \begin{cases} s_{\min} + \frac{2i}{K}(s_{\max} - s_{\min}) & \text{if } i < \frac{K}{2} \\ s_{\max} - \frac{2(K-i)}{K}(s_{\max} - s_{\min}) & \text{if } i \geq \frac{K}{2} \end{cases}, \tag{9}$$

where $s_{\min}$ and $s_{\max}$ indicate the minimum and maximum guidance scales along the temporal axis. In our experiments, we define $s_{\min}$ as 1.0 and $s_{\max}$ as 2.5. This triangle scheme assigns moderate guidance scales to the frames that will be used as conditions in the next prediction round. Due to sufficient temporal interaction, the quality of intermediate frames can also propagate to the frames that have lower guidance scales. As illustrated in Fig. 15, this technique adeptly mitigates the saturation drift problem while enhancing details. To improve perceptual continuity, we split the generated latent into clips with an overlap of 3 frames before sending them to the video-aware decoder [5]. After decoding, the overlapped frames are averaged pixel-wise.

### C.5 Human Evaluation

Recall that we ask the participants to judge side-by-side video pairs from visual quality and motion rationality. To guarantee credible responses, we provide detailed commentary for each aspect of the human evaluation. For visual quality, we let the participants focus on the consistency and harmony of the generated content. For motion rationality, we encourage the participants to pay more attention to the plausibility of the ways that ego-vehicle and other agents move, *e.g.*, whether they are following the traffic rules and exhibiting safe behaviors. For all public models we compared, we use the official checkpoints and configurations for inference without finetuning. For the models that require textual inputs [133, 144], we set the prompt as `"realistic drive view"`.

Table 4: **Reward of commands on Waymo.** The ground truth commands generally obtain higher rewards than random command inputs, suggesting that the proposed reward function can be used as a reliable indicator for command selection.

| Condition | Average Reward |
|---|---|
| GT Com. | 0.892 |
| random Com. | 0.878 (-0.014) |

Table 5: **Effect of action independence.** Without losing generality, we choose the trajectory as a representative action for evaluation. The proposed constraint expedites the learning of actions.

| Strategy | Action | Subset FVD↓ | | | |
|---|---|---|---|---|---|
| | | forth | right | left | stop |
| *w/o* A.I. | *w/o* Traj. | 163.0 | 273.9 | 428.3 | 497.1 |
| | *w/* Traj. | 138.8 | 232.9 | 368.2 | 132.3 |
| *w/* A.I. | *w/o* Traj. | 156.2 | 263.7 | 402.9 | 463.7 |
| | *w/* Traj. | 130.7 | 230.8 | 345.7 | 118.9 |

## C.6 Reward Estimation

For each condition frame and action pair, we accumulate an ensemble with size $M = 5$ to obtain a reliable uncertainty estimation. Each sample in the ensemble is inferred for 10 denoising steps as we find it is unnecessary to generate high-quality results for uncertainty estimation. The coefficient $\beta$ in the correlating strategy [35, 95] is set to 0.5.

## C.7 Ablation Studies

For the ablation of loss functions, we train each variant on OpenDV-YouTube [136] for 10K steps at a spatial resolution of 576×1024. All ablations, including the additional ablations in Appendix D, are initialized by loading the pretrained checkpoint of SVD [5] and conducted with 8 A100 GPUs.

# D Additional Experiments

## D.1 Parameter Sensitivity of Reward Estimation

To investigate how the number of denoising steps and the ensemble size influence the performance of the proposed reward function, we repeat the reward estimation procedure in Sec. 4.3 with different hyperparameter settings. We start off by using 5 denoising steps and an ensemble size of 5 for each sample. We then test two variants that double the computational cost by increasing the denoising steps to 10 (our default setup in Appendix C) and increasing the ensemble size to 10 respectively. Following Sec. 4.3, we plot the correlation of the estimated rewards with L2 errors for the three variants in Fig. 14. The results show that increasing the number of denoising steps can greatly enlarge the relative contrast of rewards, suggesting that denoising step is a more important factor than ensemble size under the same computation budget for reward estimation.

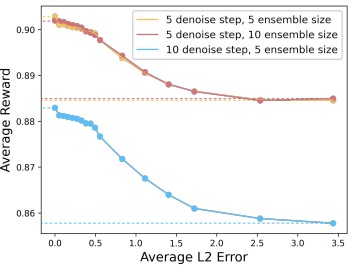

Figure 14: **Sensitivity of reward estimation to hyperparameters.** Increasing the number of denoising steps can produce more discriminative rewards, whereas increasing the ensemble size can slightly stabilize the estimations.

## D.2 Reward Estimation for Commands

To demonstrate that the proposed reward function is also applicable for other actions, we estimate the rewards of ground truth commands from Waymo and compare them with the rewards of random commands. The results in Table 4 suggest that our reward is also competent for command selection.

## D.3 Action Independence Constraint

To prove the efficacy of our learning strategy for action control, we conduct a comparison by removing the action independence constraint proposed in Sec. 3.2. We train two variants on nuScenes [10] at the resolution of 320×576 pixels for 62K steps. The comparison results are presented in Table 5.

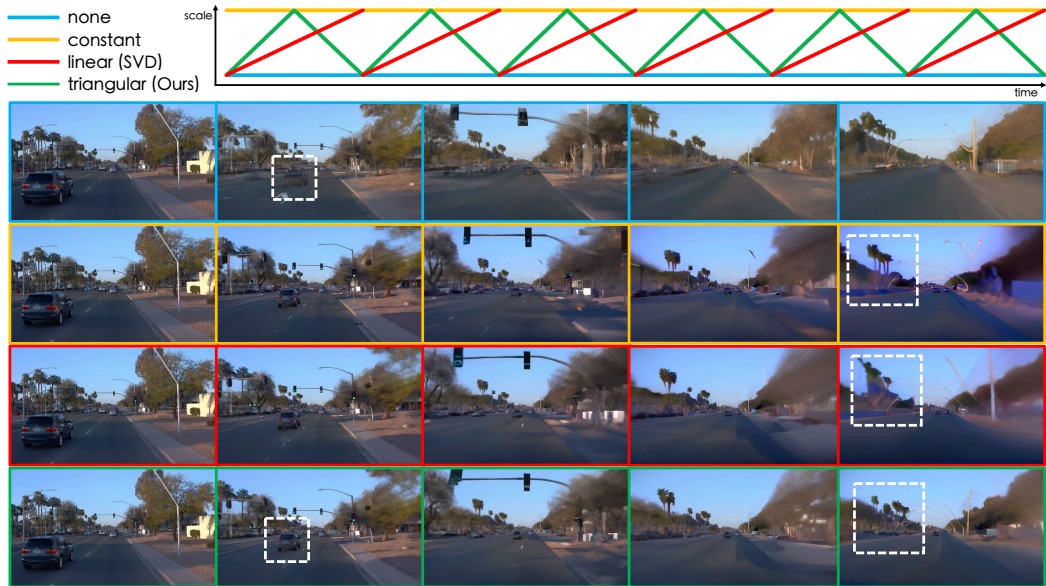

Figure 15: **Effect of guidance scale.** We predict 15s long-term videos with different CFG schemes. Our method achieves the optimal equilibrium between detail generation and saturation maintenance.

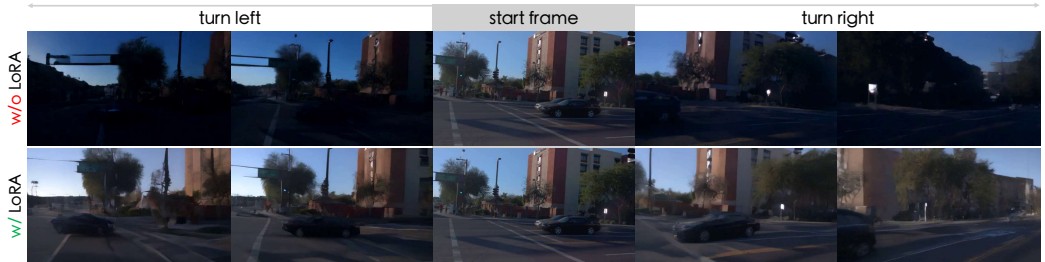

Figure 16: **Necessity of LoRA adaptation.** Training newly added projections alone without LoRA results in visual corruptions. The compared variants are trained on nuScenes and inferred on Waymo.

### D.4 Triangular Guidance Scheme

We further compare the introduced classifier-free guidance scheme with the vanilla scheme and the linear scheme [5] to verify its necessity. Fig. 15 shows that our triangular scaling attains the best trade-off between visual quality and saturation preservation.

### D.5 LoRA Adaptation

To show the necessity of applying LoRA in Sec. 3.2, we train two variants at a resolution of $320 \times 576$ pixels for 30K iterations. With the pretrained UNet weights fixed, we let one variant train LoRA and action projection layers in the attention blocks, while the other adjusts new projection layers only. As shown in Fig. 16, adding LoRA is essential for action control learning.

### D.6 Action Control Consistency

In Table 6, we report the complete FVD scores of Fig. 8, which further validates the effectiveness of all kinds of action controls. Note that since our "stop" subset consists of samples where the final displacements are within 2 meters, the goal points typically do not appear in these videos. Hence, for the experiment that uses goal point as action condition on the "stop" subset, most samples are generated in the same way as the action-free mode.

Table 6: **Complete FVD scores of different action categories.** We obtain the FVD scores on four subsets divided by command categories. All types of action controls are effective across all categories.

| Dataset | Condition | Subset FVD ↓ | | | | |
|---|---|---|---|---|---|---|
| | | forth | right | left | stop | average |
| nuScenes | action-free | 135.6 | 405.6 | 513.8 | 414.1 | 367.2 |
| | + goal point | 122.4 | 315.6 | 439.6 | 413.5 | 322.7 |
| | + command | 122.2 | 299.7 | 485.6 | 261.6 | 292.2 |
| | + angle & speed | 122.8 | 285.6 | 397.8 | 114.1 | 230.0 |
| | + trajectory | 125.2 | 229.2 | 357.7 | 118.5 | 207.6 |
| Waymo | action-free | 145.9 | 407.6 | 529.9 | 164.1 | 311.8 |
| | + command | 122.5 | 331.5 | 496.9 | 143.9 | 273.7 |
| | + trajectory | 126.3 | 285.5 | 527.6 | 136.5 | 268.9 |

## D.7 Human Evaluation with GenAD

To our best knowledge, there is not a driving-specific world model publicly available so far, making it hard to conduct qualitative human evaluation. Hence, we mainly compare Vista against the existing methods with the officially reported FID and FVD scores in Table 2.

To demonstrate the considerable improvements in visual quality and motion rationality, we conduct an extra human evaluation with the state-of-the-art GenAD model [136]. Since GenAD processes a 4-second video each time, we perform autoregressive prediction to extend Vista 's output to 5 seconds and trim the last second to align with GenAD's duration. To avoid any bias caused by resolution and frequency, we downsample the outputs of Vista (576×1024 resolution at 10 Hz) to 256×448 resolution at 2 Hz. The evaluation follows the same procedure specified in Sec. 4.1.

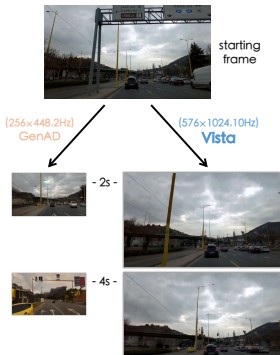

Figure 17: **Perceptual disparity between GenAD and Vista.**

We collect 25 diverse samples from the unseen OpenDV-YouTube-val set and invite 20 volunteers for evaluation. We ask the volunteers to choose the video they deem better. As a result, Vista is preferred in 94.4% and 94.8% of the time on visual quality and motion rationality respectively. This attests that Vista, in spite of experiencing a large perceptual reduction due to downsampling, still exhibits a significant advantage over GenAD in generation quality. We also compare the predictions of GenAD and Vista in Fig. 17, showing the superiority of Vista in resolution and fidelity.

## E   Additional Visualizations

### E.1   Generalization Ability

We further demonstrate the strong generalization ability of Vista by deploying it to different scenarios in the wild. The results in Fig. 18 and Fig. 19 illustrate that Vista can make high-fidelity predictions in a very diverse range of scenarios.

### E.2   Long-Horizon Prediction

In addition to Fig. 6, we provide more qualitative visualizations of long-horizon prediction in Fig. 20. Vista can continuously predict long-term futures with consistent content and motion.

### E.3   Action Controllability

We provide more prediction results with different action inputs in Fig. 21. The results on OpenDV-YouTube-val [136] and Waymo [112] show that the versatile controllability of Vista can be readily transferred to different domains in a zero-shot manner.

### E.4   Counterfactual Reasoning Ability

Counterfactual reasoning ability is one of the emergent abilities of world models [36]. As shown in Fig. 22, Vista can effectively predict the counterfactual consequences caused by abnormal actions.

### E.5 Human Evaluation Cases

To demonstrate the diversity of the scenes selected for human evaluation (Sec. 4), we show all cases gathered from OpenDV-YouTube-val [136], nuScenes [10], Waymo [112], and CODA [79] in Fig. 23.

## F  Licence of Assets

Our training and evaluation utilize the data from four publicly licensed datasets [10, 79, 112, 136]. Our implementation is based on the codebase of SVD [5], which uses the MIT license. The pretrained checkpoint of SVD is distributed under the stable video diffusion non-commercial community license.

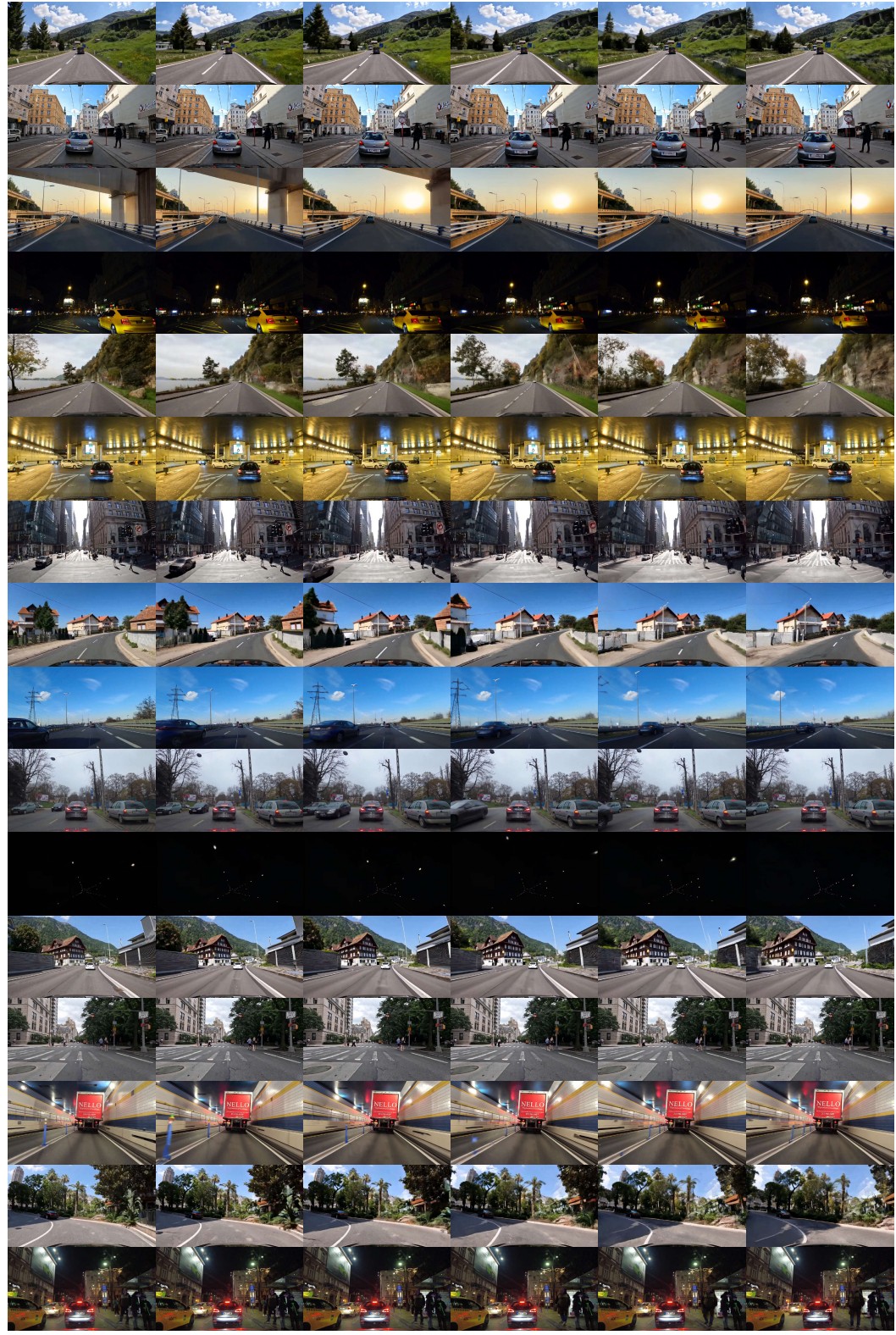

Figure 18: **Generalization ability of Vista.** We apply Vista across diverse scenes (*e.g.*, countrysides and tunnels) with unseen camera poses (*e.g.*, the perspective of a double-decker bus). Our model can predict high-resolution futures with vivid behaviors of vehicles and pedestrians, exhibiting strong generalization abilities and profound comprehension of world knowledge. Best viewed zoomed in.

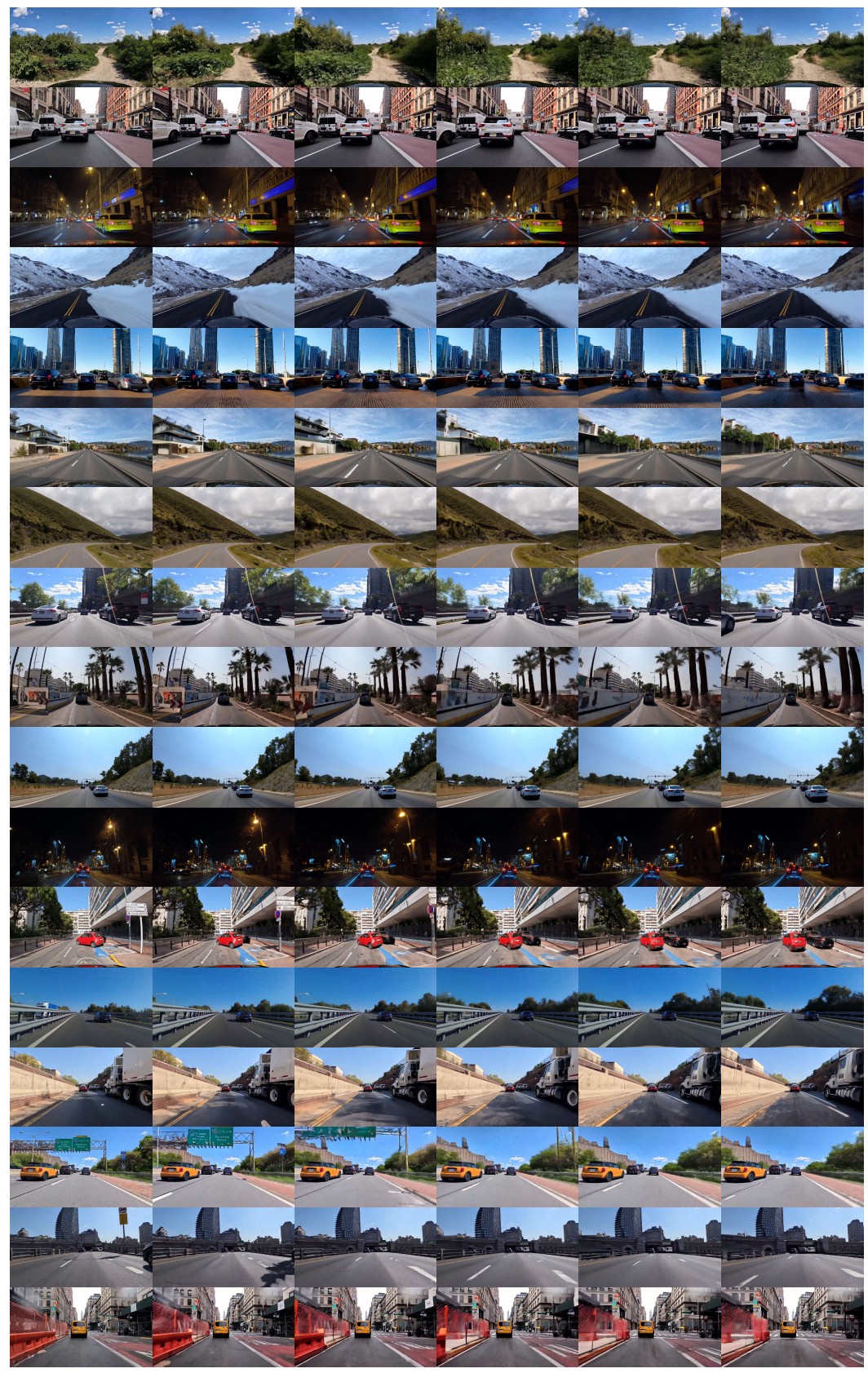

Figure 19: **Generalization ability of Vista in more scenarios.** Best viewed zoomed in.

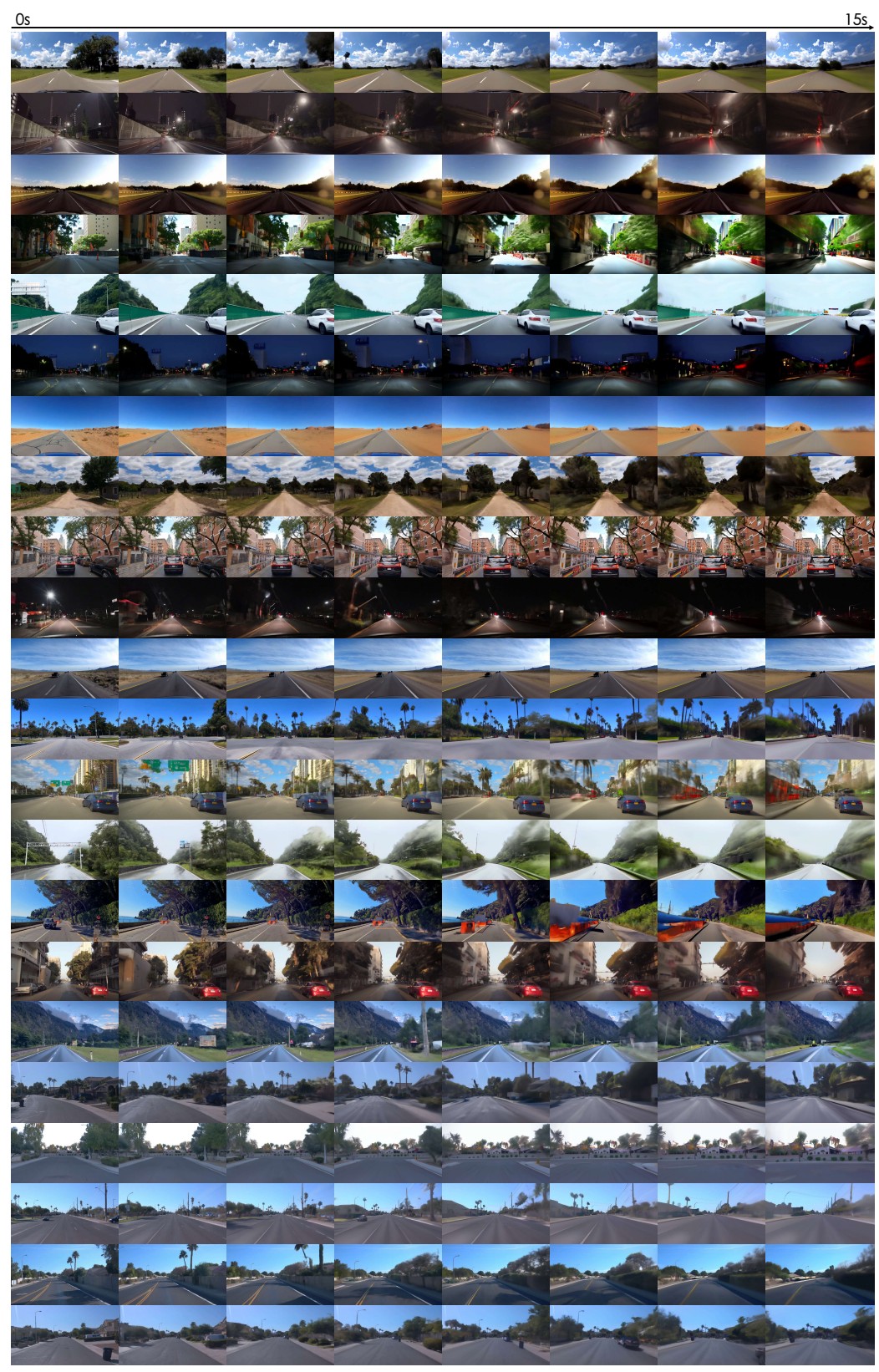

Figure 20: **Additional results of long-horizon prediction.** Our model can autoregressively simulate long driving experiences with marginal quality decline. All videos continue for 15 seconds at 10 Hz.

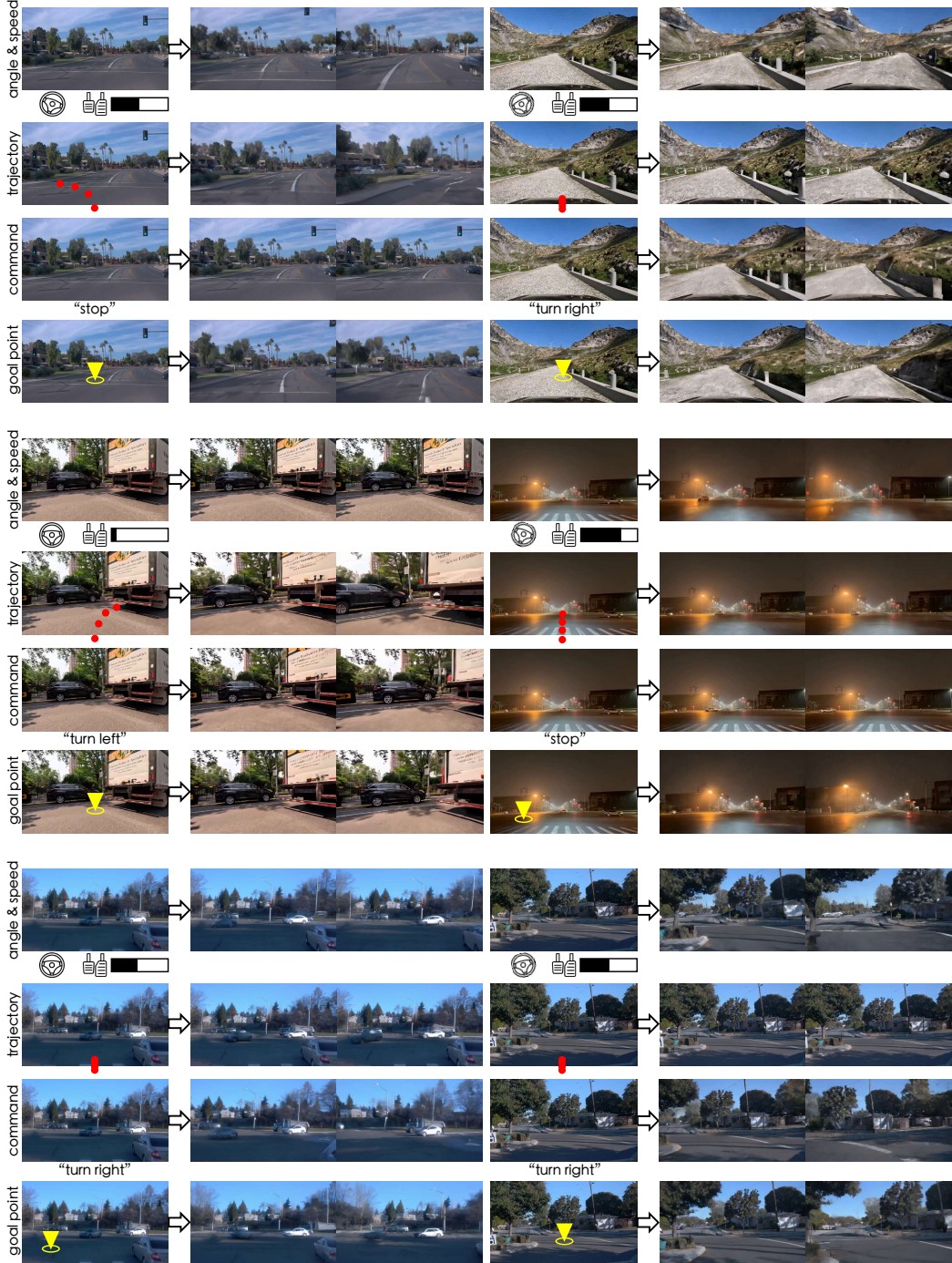

Figure 21: **Additional results of action controllability.** We trial different action conditions across multiple scenes from OpenDV-YouTube-val and Waymo. The behaviors of the ego-vehicle can be consistently controlled by various kinds of interventions.

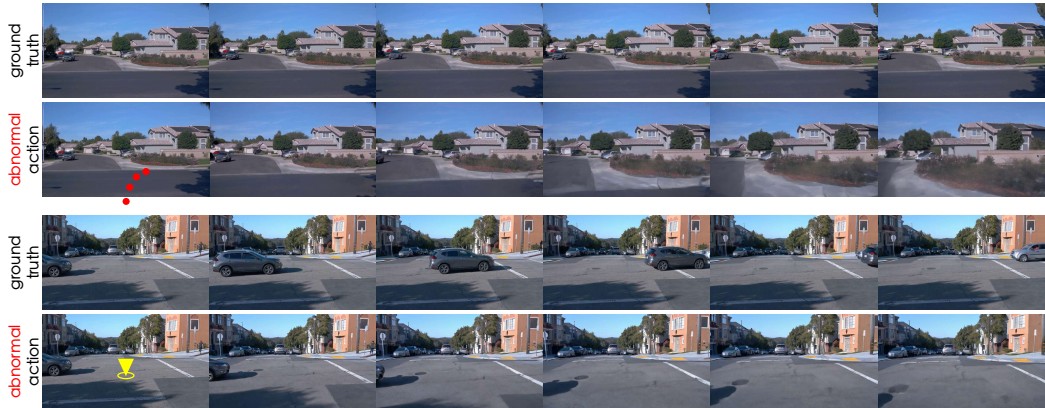

Figure 22: **Counterfactual reasoning ability.** By imposing actions that violate the traffic rules, we discover that Vista can also predict the consequences of abnormal interventions. In the first example, the ego-vehicle passes over the road boundary and rushes into the bush following our instructions. In the second example, the passing car stops and waits to avoid a collision when we force the ego-vehicle to proceed at the crossroads. This showcases Vista's potential for facilitating closed-loop simulation.

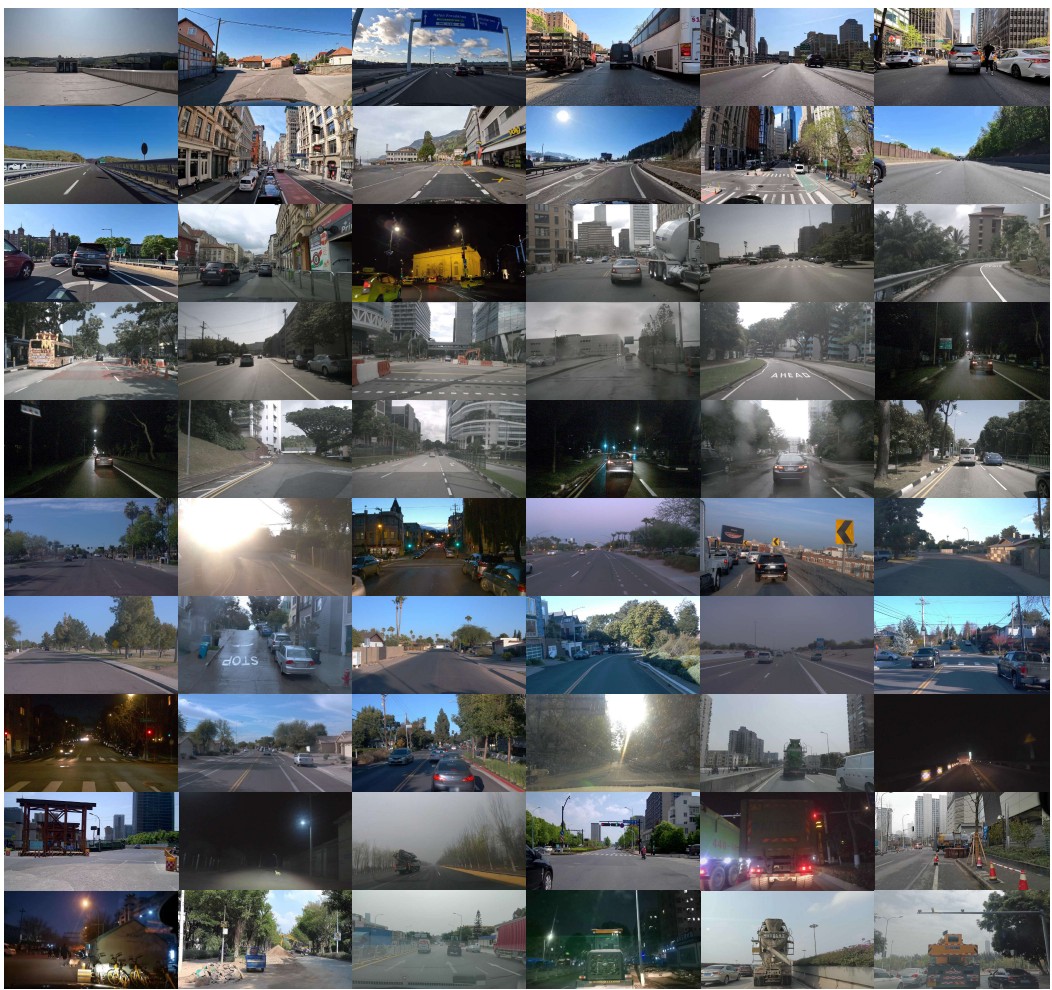

Figure 23: **Diverse scenes collected for human evaluation.** We carefully curate 60 scenes from OpenDV-YouTube-val, nuScenes, Waymo and CODA. The distinctive attributes of each dataset jointly represent the diversity of real-world environments, permitting a comprehensive human evaluation.

