# OpenReview forum: "Vista: A Generalizable Driving World Model with High Fidelity and Versatile Controllability"
_NeurIPS.cc/2024/Conference — NeurIPS 2024 poster_

### Official Review · Reviewer_bcia · 2024-06-13

**Soundness:** 2
**Presentation:** 3
**Contribution:** 2
**Rating:** 5
**Confidence:** 3

**Summary:**

This paper presents Vista, a driving world model that predict future driving video based on video diffusion model. In particular, the author introduces the idea of conditioning on prior frames and two domain-specific losses to capture dynamics and preserve structures for driving scenarios. By using LoRA, Vista also supports controllable (via e.g., trajectory, angle/speed) generation in a zero-shot manner. The author evaluates Vista by showing its high-resoluation videos, low FID on nuScenes and high human evaluation scores.

**Strengths:**

-neat domain-specific improvement on controllable driving video generation.

-the paper is clear and easy to follow.

**Weaknesses:**

-while the author discussed the differences between Vista and GenAD in Appendix A-Q6 (e.g., different controllability design, and results with higher resolution and lower FID), the improvements seem to be on the incremental side. Besides, GenAD additionally supports text command and its qualitative results of GenAD looks as good as Vista.

-the proposed conditioning on the prior frame and the two domain-specific losses and the use of LoRA while being helpful and neat, are not super novel.

-human evaluation was not conducted between Vista and driving specific generation models (e.g., GenAD).

**Questions:**

-Can you provide more discussions on why the proposed methods are very novel when there are already similar work (e.g., GenAD) on this similar task?

-Potential human evaluation results between Vista and GenAD (and/or other driving domain specific generation models)?

**Limitations:**

Yes

---

> ### Author Rebuttal · Authors · 2024-08-07
>
> Thanks for your time and effort in reviewing our paper. We provide detailed explanations below to solve the questions and some potential misunderstandings.
>
> > **W1&W2&Q1**: The improvements from GenAD seem to be incremental. The proposed dynamic priors, loss functions, and LoRA adaptation are not super novel when there are already similar works (e.g., GenAD) on this similar task.
>
> In our humble opinion, there are several fundamental differences between Vista and previous works like GenAD. Our work is a pioneering attempt to build a generalizable high-fidelity driving world model that can be controlled by multi-modal actions. To this end, we propose a series of improvements that effectively extend Vista’s abilities:
> - With our efficient training strategies, Vista has acquired versatile action controllability that can be readily generalized to unseen scenarios. In stark contrast, none of the existing works has ever enabled such ability.
> - We propose a novel reward function and validate its zero-shot efficacy in Fig. 9 and Table 3, which could serve as a potential avenue to assess actions in the wild.
> - Thanks to our innovative techniques, Vista achieves a non-marginal performance gain of **55%** in FID and **51%** in FVD compared to GenAD while being more compact in model size. We provide an intuitive comparison between Vista and GenAD in **Fig. 2 of the rebuttal PDF**. We also meticulously conduct a human evaluation between these two methods, where Vista achieves a **94.4%** win rate in Visual Quality and a **94.8%** win rate in Motion Rationality. Please refer to our answer to the next question.
> - Beyond the superior visual quality and spatiotemporal resolution, we also enable coherent long-horizon prediction ability, which is underexplored in previous works.
>
> We believe that all these contributions will shed light on future investigations in developing driving world models.
>
> > **W1&W3&Q2**: Human evaluation was not conducted between Vista and driving-specific generation models (e.g., GenAD). Qualitative results of GenAD look as good as Vista.
>
> To our best knowledge, there is not a driving-specific world model publicly available so far, making it hard to conduct qualitative human evaluation. Therefore, we mainly compare Vista against the existing methods with the officially reported FID and FVD scores in our paper.
>
> To demonstrate the considerable improvements in visual quality and motion rationality, we conduct a human evaluation with the state-of-the-art GenAD model *as requested*. We follow the two-stage training strategy and use the same training compute as specified in their original paper. Since GenAD processes a 4-second video each time, we perform autoregressive prediction to extend Vista’s output to 5 seconds and trim the last second to align with GenAD’s duration. To avoid any bias caused by resolution and frequency, we also downsample the outputs of Vista (576x1024 resolution at 10 Hz) to 256x448 resolution at 2 Hz for fairness. The human evaluation is conducted following the same procedure in Sec. 4.1.
>
> During the rebuttal period, we collect 25 diverse samples from the unseen OpenDV-YouTube-val set and invite 20 volunteers for evaluation. We ask the volunteers to choose the video they deemed better. As a result, Vista is preferred in **94.4%** and **94.8%** of the time on Visual Quality and Motion Rationality respectively. This certifies that Vista, even when downsampled to a much lower spatiotemporal resolution (which is a significant perceptual reduction), has a remarkable advantage over GenAD in generation quality. We also append a qualitative comparison between GenAD and Vista in **Fig. 2 of the rebuttal PDF**, showing the superiority of Vista in resolution and quality.
>
> > **W1**: Vista does not support text commands while GenAD does.
>
> Although GenAD provides a control interface for text commands, its advantage over Vista is limited for three main reasons:
> - The text commands used by GenAD are annotated by first classifying videos into discrete categories and then mapping to text templates from a predefined dictionary. Thus, the text commands used in GenAD and categorical embeddings learned by Vista are **functionally similar**.
> - As stated in the limitations of the GenAD paper, the auto-labeling process of text annotations may incur conflicts with ego intentions. Unlike GenAD that requires labeling all videos from YouTube, we explore a collaborative training strategy that allows learning open-world action controllability from the public dataset with reliable annotations. This strategy circumvents the labor of auto-labeling that may cause accumulated errors and learning instability.
> - Text commands are often ambiguous in expressing precise controls. Therefore, in Vista, we focus more on versatile action controllability ranging from high-level intentions to low-level maneuvers, empowering the applications for various purposes.

---

> > ### Comment · Reviewer_bcia · 2024-08-11
> > **Thank you for the response**
> >
> > Thank you for your detailed response which mostly addresses my concerns and I will increase my initial rating.

---

> > > ### Author Response · Authors · 2024-08-11
> > >
> > > Thank you for the kind response! We would appreciate specification on any concerns you may have, which will allow us to provide further information. Thank you.

---

### Official Review · Reviewer_ApTu · 2024-07-10

**Soundness:** 4
**Presentation:** 4
**Contribution:** 3
**Rating:** 7
**Confidence:** 5

**Summary:**

Vista is a generalizable driving world model that excels in high fidelity and versatile controllability. By introducing novel losses to enhance the learning of dynamics and structural information, and integrating a unified conditioning interface for diverse action controls, Vista achieves high-resolution predictions and adapts seamlessly to various scenarios in a zero-shot manner. Extensive experiments demonstrate that Vista outperforms state-of-the-art video generators and driving models, showcasing significant improvements in prediction fidelity and action evaluation capabilities​.

**Strengths:**

1. High Fidelity Predictions: Vista achieves accurate and realistic future predictions at high resolutions.
2. Versatile Action Controllability: Supports diverse control inputs from high-level commands to low-level maneuvers.
3. Strong Generalization: Seamlessly adapts to diverse and unseen environments in a zero-shot manner.
4. Superior Performance: Outperforms state-of-the-art models in prediction fidelity and evaluation metrics.

**Weaknesses:**

1. I have some questions regarding the production of Figure 5. Is the SVD in Figure 5 used as is, or has it been retrained? Are there any other action control modules? Additionally, while the long-term generation of Vista seems consistent, most of the details are quite blurry. I am also confused about the point that "the prediction of SVD does not commence from the condition image."

2. Regarding the misalignment in DynamiCrafter in Figure 4, it is actually because during training of DynamiCrafter , the condition frame is not always the first frame, but is randomly extracted from the video. Therefore, you can see that the fourth column of DynamiCrafter is consistent with the input frame. This is not a misalignment.

I will raise my rating if all of my concerns are well addressed.

**Questions:**

Mentioned in the weakness section

**Limitations:**

Yes.

---

> ### Author Rebuttal · Authors · 2024-08-07
>
> Thank you for the thoughtful comments and questions. We answer each question below and will incorporate all feedback in the revision.
>
> > **W1**: Details related to the production of Fig. 5. (1) Has SVD been retrained? Are there any other action control modules? (2) The details of long-term generation are blurry. (3) The meaning of "the prediction of SVD does not commence from the condition image".
>
> (1) We did not retrain SVD for this comparison. The results of SVD in Fig. 5 are generated using the official checkpoint and codebase without any modification. The samples in Fig. 5 are all action-free predictions without action conditioning, thus no action control modules are needed for SVD here.
>
> (2) It is possible that the long-horizon rollouts may result in degradations of details. In fact, long-term prediction remains a challenge in this research direction. To overcome this challenge, we have explored some techniques that optimize the fidelity of long-term prediction (e.g., triangular classifier-free guidance in Appendix C.4). Moreover, although there is still room for improvement, it is noteworthy that our method has significantly outperformed the existing methods, with nine times more frames than Drive-WM and much better content consistency compared to SVD. As discussed in Appendix A-Q7, we will continue exploring solutions for long-term fidelity in future works.
>
> (3) We apologize for the confusion. We use "the prediction of SVD does not commence from the condition image" to express that the first frame predicted by SVD is not identical to the condition image. This misalignment prevents SVD from performing autoregressive long-term rollout, as the consecutive clips predicted by SVD are not consistent in content. We will clarify this in the revision.
>
> > **W2**: The misalignment in DynamiCrafter is because of its training settings.
>
> Thanks for the comment. We agree that the misalignment of DynamiCrafter is due to its training settings, where a random frame is sampled as the condition image. Since DynamiCrafter is a prominent general-purpose video generator, we compare our model to DynamiCrafter to demonstrate the better fidelity of Vista and its distinctions in capabilities. Unlike existing general-purpose video generators for content creation, Vista is designed to make plausible future predictions while preserving high quality. We will clarify this in our revision to avoid confusion.

---

> > ### Comment · Reviewer_ApTu · 2024-08-11
> > **Response**
> >
> > All of my concerns are well addressed. I will increase my score to accept

---

> > > ### Author Response · Authors · 2024-08-12
> > >
> > > Thank you for your response.  We will integrate your advice in our revision.

---

### Official Review · Reviewer_Fp7V · 2024-07-12

**Soundness:** 3
**Presentation:** 4
**Contribution:** 4
**Rating:** 7
**Confidence:** 4

**Summary:**

The paper presents a method named Vista, a novel driving world model that addresses limitations in generalization, prediction fidelity, and action controllability. Key contributions include introducing novel loss functions for high-resolution prediction, a latent replacement approach for coherent long-term rollouts, and versatile action controls ranging from high-level commands to low-level maneuvers. Vista demonstrates strong generalization in zero-shot settings and establishes a generalizable reward function for evaluating driving actions. Extensive experiments on multiple datasets demonstrate that Vista outperforms advanced general-purpose video generators in over 70% of comparisons, surpasses the best-performing driving world model by 55% in FID and 27% in FVD, and establishes a generalizable reward function for real-world driving action evaluation.

**Strengths:**

1. The paper introduces a novel approach to driving world models by incorporating high-fidelity predictions and versatile action controllability. This combination addresses existing gaps in generalization, prediction fidelity, and action flexibility, representing a significant step forward in autonomous driving research.
2. The paper presents a well-designed methodology that systematically addresses the limitations of existing driving world models. The integration of dynamic prior injection and versatile control mechanisms is methodologically sound and effectively implemented.
3. The paper is well-structured, with clear and logical sections that guide the reader through the problem formulation, methodology, experiments, and conclusions. Figures and tables are used effectively to illustrate key points and results.

**Weaknesses:**

1. Vista seems cannot to generate surround-view video, this limitation may restrict the method's effectiveness and generalizability in real-world scenarios where a comprehensive 360-degree view is crucial.
2. The closed-loop evaluation is demonstrated through only a few cases, raising concerns about the robustness and reliability of integrating the closed-loop process with the generative model. Expanding the evaluation to include a broader range of scenarios and detailed performance metrics would help assess the seamless integration of the closed-loop driving process with the generative model.
3. The paper introduces dynamic prior injection for coherent long-term rollouts, but the implementation details and impact of this component are not sufficiently elaborated. Providing more detailed information on this mechanism, including its implementation, theoretical basis, and specific impact on long-term prediction consistency, would enhance understanding.
4. The paper lacks a detailed analysis of the computational resources required for training and inference using the Vista framework. The potentially high computational demands could limit the scalability and real-time applicability of the approach.

**Questions:**

1. Can the authors provide more details on the implementation of the dynamic prior injection mechanism? How does it theoretically support coherent long-term rollouts?
2. Provide additional experimental results demonstrating the model's performance in zero-shot settings across diverse and unseen environments. This would help validate the framework’s generalization capabilities and robustness in real-world applications.
3. Expanding the Vista framework to support surround-view video generation could significantly enhance its applicability and robustness in real-world autonomous driving scenarios, as a comprehensive 360-degree view is crucial for effective navigation and situational awareness. Additionally, extending the closed-loop evaluation to include a broader range of scenarios and detailed performance metrics would provide a more thorough assessment of the model's robustness and reliability. Such enhancements would further solidify the contributions of this work and pave the way for future advancements. Nonetheless, the current contributions are commendable, and I look forward to seeing future developments in this area.

**Limitations:**

please refer to weaknesses and questions.

---

> ### Author Rebuttal · Authors · 2024-08-07
>
> Thanks for your insightful and positive feedback. The following are our responses.
>
> > **W1&Q3**: Surround-view generation is not supported.
>
> We agree that supporting surround-view generation would further help driving. We are planning to extend Vista to multi-view settings like Drive-WM (Wang, et al.) in our future works.
>
> In this paper, we focus on the front-view setting for three main reasons:
> - The front view setting allows leveraging diverse data sources (e.g., the worldwide OpenDV dataset). Conversely, the distinctions in multi-view videos from various datasets, such as different numbers of cameras, hinder unified modeling and data scaling.
> - Models that focus on the front view can be seamlessly applied to different datasets without adaptation (e.g., DriveLM (Sima, et al.)), broadening their applicability across datasets.
> - Though incomplete, the front view often contains most of the information necessary for driving. As demonstrated in NAVSIM (Dauner, et al.), using the front view alone results in only a 1.1% performance drop in collision rate compared to using five surround-view cameras.
>
> > **W2&Q3**: Evaluating the closed-loop process in a broader range of scenarios and detailed metrics.
>
> From our understanding, the closed-loop process here refers to controlling Vista to create a closed-loop simulation.
>
> To address the concern, we introduce an additional metric, *Trajectory Difference*, to assess the control consistency. Following GenAD, we train an inverse dynamics model (IDM) that estimates the corresponding trajectory from a video. We then send Vista's prediction to the IDM and calculate the L2 difference between the ground truth trajectory and the estimated trajectory. The lower the difference, the better the control consistency Vista has. We conduct the experiments on nuScenes and Waymo (unseen by Vista). As reported in **Table 1 of the rebuttal PDF**, Vista can be effectively controlled by different types of actions, yielding more consistent motions to the ground truth.
>
> We also provide the complete FVD scores of Fig. 7 in **Table 2 of the rebuttal PDF**, which further validates the efficacy of all kinds of action controls.
>
> For the coverage of scenarios, we have shown multiple open-world samples on the anonymous demo page at the time of submission. We will provide more demonstrations in the revision. In addition, we will fully open-source our code and model to the community for free trials.
>
> > **W3&Q1**: More details on the implementation, theoretical basis, and impact of dynamic prior injection.
>
> **Implementation details**: Conventional video diffusion methods (e.g., SVD) process a sequence of noisy frames to generate a video. However, this approach cannot ensure content and motion consistency between clips in long-horizon rollouts. To address this, we inject previous frames as priors to derive the necessary information for consistent rollouts. As illustrated in Fig. 2 [Left], these dynamic priors are injected by replacing the noisy frames with the previously known frames throughout the entire denoising process. The dynamic priors are then propagated through the temporal interactions within the model. The number of dynamic priors corresponds to the number of the injected frames. To indicate the presence of dynamic priors that do not require denoising, we assign different timestep embeddings to these frames. In our implementation, we create a frame-wise mask to uniformly allocate the dynamic priors and the timestep embeddings. We will refine this part accordingly in the revision.
>
> **Theoretical basis**: As discussed in Appendix A-Q1, it is necessary to input sufficient information so that the model can learn to derive position, velocity, and acceleration for coherent future prediction. For example, without knowing acceleration, the model cannot determine whether another car in view is moving faster or slower. Such uncertainty will result in unnatural motions with respect to the historical frames. To fully obtain these priors, at least three consecutive frames with the same interval are required. To ensure temporal consistency while predicting as many frames as possible each time, we always use three previous frames as dynamic priors during long-horizon rollouts.
>
> **Impact evaluation**: To further demonstrate the effectiveness of dynamic priors, we conduct a quantitative evaluation in **Table 1 of the rebuttal PDF**. Specifically, we use the inverse dynamics model to infer the trajectories of the predicted videos with different orders of dynamic priors. Extensive results show that increasing the order of dynamic priors can consistently improve the coherence to the ground truth motion.
>
> > **W4**: A detailed analysis of the computational resources required for training and inference.
>
> We have provided a thorough description of the training in Appendix C.3. The entire training process takes about two weeks, with the first phase taking one week on 128 A100 GPUs and the second phase taking another week on 8 A100 GPUs. For inference cost, it takes 70-80 seconds on a single A100 to predict 25 frames. Note that the inference could be greatly accelerated using some well-established techniques as discussed in Appendix A-Q7. While this is not in the scope of this paper, we will explore these techniques for downstream applications in the future.
>
> > **Q2**: Additional experimental results in zero-shot settings across diverse and unseen environments.
>
> All qualitative visualizations in our paper are produced under the zero-shot setting in open-world scenarios. Moreover, except for the results on nuScenes, all quantitative results are demonstrating Vista’s zero-shot performance (quality, duration, controllability, etc.). As described in Sec. 4.1, our experiments involve zero-shot evaluation on three unseen/geofenced datasets (OpenDV-YouTube-val, Waymo, CODA). Fig. 20 also shows the environmental diversity of our human evaluation. We will specify the zero-shot settings in the revision.

---

> > ### Comment · Reviewer_Fp7V · 2024-08-11
> >
> > Thank you to the authors for your detailed response, and I think most of my concerns are almost resolved. I want to keep my rating unchanged.

---

> > > ### Author Response · Authors · 2024-08-11
> > >
> > > Thanks for responding to our feedback and recognizing our contributions. We really appreciate your help in improving our work!

---

### Author Rebuttal · Authors · 2024-08-07

Dear reviewers and ACs:

We express our sincere gratitude to the reviewers for their thorough and constructive comments. It is encouraging that all reviewers have acknowledged our pioneering efforts in establishing a driving world model with versatile controllability.

We have carefully taken each comment into consideration. The attached **rebuttal PDF** includes two tables with quantitative results and two figures for illustration. Please refer to other rebuttal modules below for our detailed responses to each comment. We will integrate these results and discussions into our revised paper.

We hope that our rebuttal can address the concerns you may have. You are more than welcome to ask any further questions. We are looking forward to your feedback!

Best regards,
Authors of Submission574

---

### Decision · Program_Chairs · 2024-09-25

**Decision:**

Accept (poster)

**Comment:**

Overall, the reviewers have positively recommended the paper. The ACs agree with the reviewers' comments.
Here are some additional comments for the authors to potentially improve the quality of the final camera-ready version:
	1	Quantitative evaluation could be more comprehensive -> they only show evaluation on nuScenes val set
	2	No mention of the baseline models used for the human evaluation study — they tested on waymo and coda so which models were used for that?
	3	Most of the videos/demos on the website are of empty streets. How good is the model in predicting motion of other vehicles/objects in the scene?
	4	For the new losses, we only see two examples that motivated the use of the new losses (Fig 12). That might not be very comprehensive. Maybe quantitative evaluation showing how much these losses affect the performance?
	5	They introduce different control techniques but they don’t use them simultaneously. It would be interesting to see an experiment with multiple controls used together. Would that be too strong of a signal for the generations? How could these controls interact together?

Unanswered questions:
	1	Which resolution is used for the quantitative evaluation ? Are the metrics much better for Vista because of the resolution?
	2	The controls are mainly used from NuScenes because it’s the only dataset with these annotations. Why not trying unsupervised/semi-supervised methods to get these annotations for more data? Would that affect the controllability ?